# Next-generation proteomics for quantitative Jumbophage-bacteria interaction mapping

Andrea Fossati [1,2,3,6], Deepto Mozumdar[4,6], Claire Kokontis[4], Melissa Mèndez-Moran [5], Eliza Niewieglowska [5], Adrian Pelin[1,2,3], Yuping Li [4], Baron Guo[4], Nevan J. Krogan [1,2,3], David A. Agard [5], Joseph Bondy-Denomy [4] ✉ & Danielle L. Swaney [1,2,3] ✉

Host-pathogen interactions are pivotal in regulating establishment, progression, and outcome of an infection. While affinity-purification mass spectrometry has become instrumental in characterizing such interactions, it suffers from limitations in scalability and biological authenticity. Here we present the use of co-fractionation mass spectrometry for high throughput analysis of host-pathogen interactions from native viral infections of two jumbophages ($\phi$KZ and $\phi$PA3) in *Pseudomonas aeruginosa*. This approach enabled the detection of > 6000 unique host-pathogen interactions for each phage, encompassing > 50% of their respective proteomes. This deep coverage provided evidence for interactions between KZ-like phage proteins and the host ribosome, and revealed protein complexes for previously undescribed phage ORFs, including a $\phi$PA3 complex showing strong structural and sequence similarity to $\phi$KZ non-virion RNA polymerase. Interactome-wide comparison across phages showed similar perturbed protein interactions suggesting fundamentally conserved mechanisms of phage predation within the KZ-like phage family. To enable accessibility to this data, we developed PhageMAP, an online resource for network query, visualization, and interaction prediction (https://phagemap.ucsf.edu/). We anticipate this study will lay the foundation for the application of co-fractionation mass spectrometry for the scalable profiling of host-pathogen interactomes and protein complex dynamics upon infection.

Protein–protein interactions (PPIs) are the fundamental building blocks of cellular complexity, and their perturbation and rewiring have profound effects on the proteome and cell fate. During an infection, the interactions between host and pathogen proteome are pivotal in regulating pathogen tropism, infection progression, and, ultimately, infection outcome. Host–pathogen interaction (HPI) mapping using affinity-purification mass spectrometry (AP-MS) has been instrumental in identifying host-targeted processes[1–5] and, recently, in predicting potential therapeutic targets during the SARS-CoV-2 pandemic[6–8].

Despite the successes of AP-MS for mapping HPIs, the exogenous expression and purification of individual pathogen proteins limit our ability to characterize HPIs under native expression levels and quantify

[1]J. David Gladstone Institutes, San Francisco 94158 CA, USA. [2]Quantitative Biosciences Institute (QBI), University of California San Francisco, San Francisco 94158 CA, USA. [3]Department of Cellular and Molecular Pharmacology, University of California San Francisco, San Francisco 94158 CA, USA. [4]Department of Immunology and Microbiology, University of California San Francisco, San Francisco 94158 CA, USA. [5]Department of Biochemistry, University of California San Francisco, San Francisco 94143 CA, USA. [6]These authors contributed equally: Andrea Fossati, Deepto Mozumdar. ✉e-mail: Joseph.Bondy-Denomy@ucsf.edu; danielle.swaney@ucsf.edu

how these interactions are regulated in the context of the full pathogen protein repertoire during infection as well as precluding the detection of downstream rearrangements in protein complexes beyond the viral protein of interest. While some of these limitations have been partially overcome by the introduction of endogenous tagging within the viral genome, this approach has been mostly limited to small viruses[9]. Both endogenous tagging and ectopic expression are labor-intensive processes that require the generation of numerous plasmids and hundreds or thousands of individual purifications to comprehensively probe protein-protein interactions for an entire viral proteome. This limits the scalability of AP-MS for the characterization of HPIs for larger viruses or bacteria which express hundreds or thousands of proteins.

As a result, small eukaryotic viruses have been prioritized in HPI studies[6,10], thus, extensive knowledge on interactions between larger prokaryotic viruses (bacteriophages) and their host is currently missing. This class of bacterial viruses holds great potential for the treatment of multi-drug resistant bacteria, which have increasingly been reported in the last two decades[11]. However, without a thorough understanding of putative interactions and functions of the phage gene products, it will be challenging to inform the rational design of the next generation of phage therapeutics.

To bridge this gap, here we have applied co-fractionation mass spectrometry using size-exclusion chromatography, coupled with fast data-independent acquisition MS (SEC DIA-MS)[12]. In this technique, protein complexes extracted from a native lysate are size-fractionated, and each fraction is analyzed and acquired via mass-spectrometry, resulting in a large matrix of protein intensities over molecular mass. These protein profiles are then utilized as a proxy for the assembly state under the assumption that proteins having identical peak shapes and positions were physically associated at the separation stage. Using SEC-DIA-MS, we generated two systematic phage–bacteria interactomes and measured host PPI rewiring upon phage infection in *Pseudomonas aeruginosa*. This was done for two KZ-like phages (ϕKZ[13] and ϕPA3[14]), which share 84% nucleotide sequence identity. Both are archetype Jumbophages that possess large genomes (>300 genes), with very limited organization of genes by function, hence lacking synteny. Unique to this family of phages is the presence of a large proteinaceous shell acting analogous to the eukaryotic nucleus, thus decoupling transcription from translation. This structure confers resistance to several bacterial antiphage systems such as CRISPR[15,16] and has a fundamental role in infection establishment[17] and virion production[18]. Through the prediction of PPIs using deep learning and structural modeling, we derived system-level maps of Jumbophage infection encompassing a large fraction of the phage and bacterial host proteome. These HPI maps substantially extend previous knowledge on Jumbophage predation and demonstrate the application of co-fractionation mass spectrometry for HPI profiling.

## Results

### A cross-phage study of the viral infection cycle
To understand HPIs that mediate phage infection, we infected *Pseudomonas aeruginosa* (strain PAO1) with either the ϕKZ or ϕPA3 bacteriophage for 60 min in a biological duplicate. To control for virion protein complexes (i.e., complexes present within the phage itself), parallel experiments were also performed using a mutant PAO1 strain, dubbed 'PAO1 control', that emerged under phage selection (KZ resistant mutant)[19] that resists infection from both phages as shown in Supplementary Fig. S1 (Fig. 1A). This strain expresses significantly lower levels of FliC protein (the major structural unit of the flagellum), a known receptor for ϕKZ[19].

Infected cell lysates were fractionated by size-exclusion chromatography, and each fraction (*n* = 72) was analyzed using data-independent acquisition MS (DIA-MS) coupled to high-throughput liquid chromatography[20]. To predict HPIs, we used a modified version

of the PCprophet toolkit[12,21], where the random forest classifier was replaced with a deep neural network that was trained for PPI prediction using >10 million interactions from various co-fractionation experiments[22]. Following data processing and replicate integration, we utilized deep learning to predict co-eluting (i.e., interacting) proteins based on their intensity profiles across all measured fractions for a particular condition. To further increase our confidence, we utilized two filters: first, a target-decoy approach was employed to control for randomly coeluting proteins[21], and then PPIs were filtered to those with a prediction probability of ≥0.75, resulting in a PPI false-discovery rate of less than 5%.

Derived HPI networks have been organized into a user-friendly website, PhageMAP, where users can query proteins of interest to visualize coelution patterns, interactomes, investigate different assembly states of the PAO1 proteome upon phage infection, and export their findings as publication-quality networks or coelution plots (Fig. 1B).

This experimental workflow resulted in the high-throughput and comprehensive coverage of both the bacterial and the phage proteomes. Specifically, we detected 3782 PAO1 proteins, covering 83% of the validated SwissProt entries (i.e., proteins for which experimental evidence of their existence is available) for the *Pseudomonas* pan-proteome, and 67% of the unreviewed entries (Fig. 1C). Likewise, we detected 280 proteins for ϕKZ and 198 proteins for ϕPA3, covering 75% and 53% of their proteomes, respectively (Fig. 1D).

To test the achievable robustness and resolution of our workflow, we used two benchmarks. First, the robustness of fractionation was assessed by the Pearson $R^2$ between the two replicates of a given condition. Each condition showed an average correlation of >0.8 (Fig. 1E), indicating high reproducibility in both phage infection and SEC fractionation, with most of the SEC-profile peaks overlapping within 1–2 fractions (<0.250 μL). To test the resolution achievable with our chromatographic separations, we calculated the number of SEC peaks per protein, which is a direct proxy for how many different complex assemblies a protein participates in. Approximately 45% of the identified proteins were detected in a single SEC-peak in each condition employed (Fig. 1F). While the presence of a single peak can represent detection of only a monomeric protein, we found the majority of these single-peak proteins (90/137 for ϕKZ, 75/110 for ϕPA3 and 1843/2382 for the PAO1 control) are not at their predicted monomeric molecular weight (Supplementary Fig. S2). This suggests that the protein complex assembly state of the PAO1 proteome was preserved during sample preparation and SEC fractionation.

### A high-quality interaction dataset for bacterial protein complexes
Next, we sought to investigate the recovery of known protein complexes by leveraging the partial conservation of core molecular assemblies between *P. aeruginosa* and other bacteria, such as *Escherichia coli*, for which protein complexes are more extensively annotated[23]. To visualize our data, we utilized the KZ-resistant mutant dataset and projected it using t-SNE (Fig. 2A). In this dimensionality reduction approach, neighboring points in the embedded space are derived from proteins sharing similar protein profiles, while distinct profiles results in distant points. However, due to the non-linear nature of the t-SNE algorithm, the distance between clusters and the shape of the global or local embedding cannot be interpreted back to the input data. Smaller enzymes, such as metabolic enzymes, are usually co-expressed within the same operon[24] and have been reported to dimerize or multimerize. In line with this, we observed enzymes such as the pyruvate dehydrogenase complex (Fig. 2B) and the oxoglutarate dehydrogenase complex (Fig. 2C), which migrated at an estimated MW of ≈3.5 × 10⁶ Da (expected MW ≈3.75 × 10⁶ Da) and ≈2.4 × 10⁶ Da, respectively. It is important to point that that the molecular weight estimation for these large assemblies is subject to error due to these

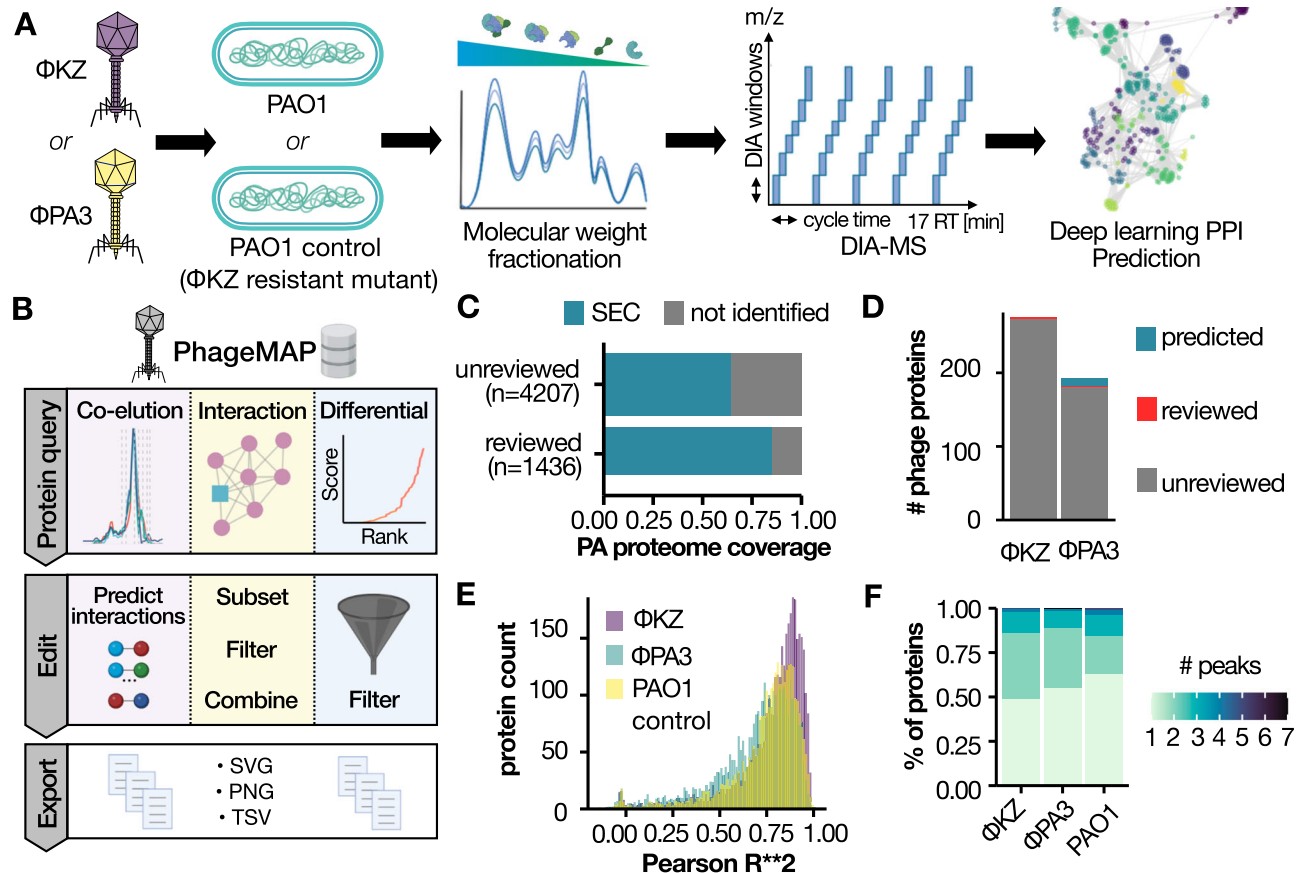

**Fig. 1 | High-throughput interaction proteomics for deep host–pathogen interaction mapping. A** SEC-MS workflow and experimental design. **B** Overview of PhageMAP analysis and workflows. **C** Recovery of *Pseudomonas* proteome by SEC-MS. **D** Barplot representing the number of phage proteins identified. **E** Correlation for all proteins identified in the experiment (*n* = 4132) between the two biological replicates. **F** Fractional distribution of the number of SEC peaks across the various phages and host.

peaks being outside the external calibration curve. To achieve MW estimation, we included in the calibration curve a pure SEC-separated 70S ribosome (Supplementary Fig. S3).

Our sample preparation also preserved membrane-bound complexes. As an example, the AAA protease complex, formed by four hexamers of the AAA protease (ftsH) and 12 copies of each single-pass membrane protein (HflK and HflC)[25], was recovered at high molecular weight in a broad peak, as shown in Fig. 2D. The large molecular weight range and sensitivity covered by our separation approach were also demonstrated in the recovery of more transient complexes such as the DNA polymerase III (dnaA, dnaE, and dnaQ) loaded with the γ complex (holA and dnaX) which plays a key role at the replication fork[26] (Fig. 2E). Finally, heterodimeric complexes such as the succinyl-coA synthetase were also recovered as demonstrated by the coelution plot in (Fig. 2F). Our manual inspections further confirm that prior knowledge can be easily incorporated into SEC-MS data analysis and allows for straightforward identification of protein complexes.

### Comparison of host-targeted processes reveals conserved and divergent predation mechanisms

After having demonstrated the proteome depth achieved in our SEC-MS dataset and the recovery of known complexes, we turned our attention to how Jumbophages re-wire *P. aeruginosa* protein complexes by evaluating differences in SEC profiles upon phage infection. Variation in SEC profiles between conditions can arise from differential assembly state (i.e., a protein profile shifting to higher or lower molecular weight), different stoichiometry within a complex, or global alterations in protein abundance.

To quantify these different cases, we employed a previously described Bayesian analysis module from the PCprophet package[12] to derive marginal likelihoods (SEC differential score) of protein-level SEC changes between φKZ and φPA3 versus the receptorless infected samples (i.e., PAO1 control). Differential analysis of two SEC profiles using PCprophet provides the SEC differential score, which represents the variation of complex intensity (stoichiometry) or peak position (assembly state). Comparing the SEC-profile differences between phage-infected PAO1 and PAO1 control revealed approximately 600 proteins showing SEC variation upon infection by either phage (Fig. 3A). Notably, there is substantial consistency in which *P. aeruginosa* proteins are altered between both phages and the degree of change in their individual SEC profiles (Fig. 3B, cor = 0.677), potentially pointing towards common pathways and complexes hijacked by φPA3 and φKZ for successful predation. When compared to an independent whole-cell lysate-proteome protein abundance measurement of the same cell lysate, we find that most of the changes at the assembly state level do not have a corresponding variation in protein abundance at the global proteome level. Altogether this observation suggests that SEC-MS offers an orthogonal view on the effect of perturbations, such as infection, on the proteome (Fig. 3C).

To identify conserved KZ-like jumbophage manipulation of the host interactome, we mapped the SEC-derived PAO1 interaction network (Fig. 3D) with the correspondent protein-level differential data derived from the comparison between phage and uninfected samples. Although a large portion of the nodes do not have a functional annotation, we identified several functional classes where their components were significantly altered upon Jumbophage phage infection, as

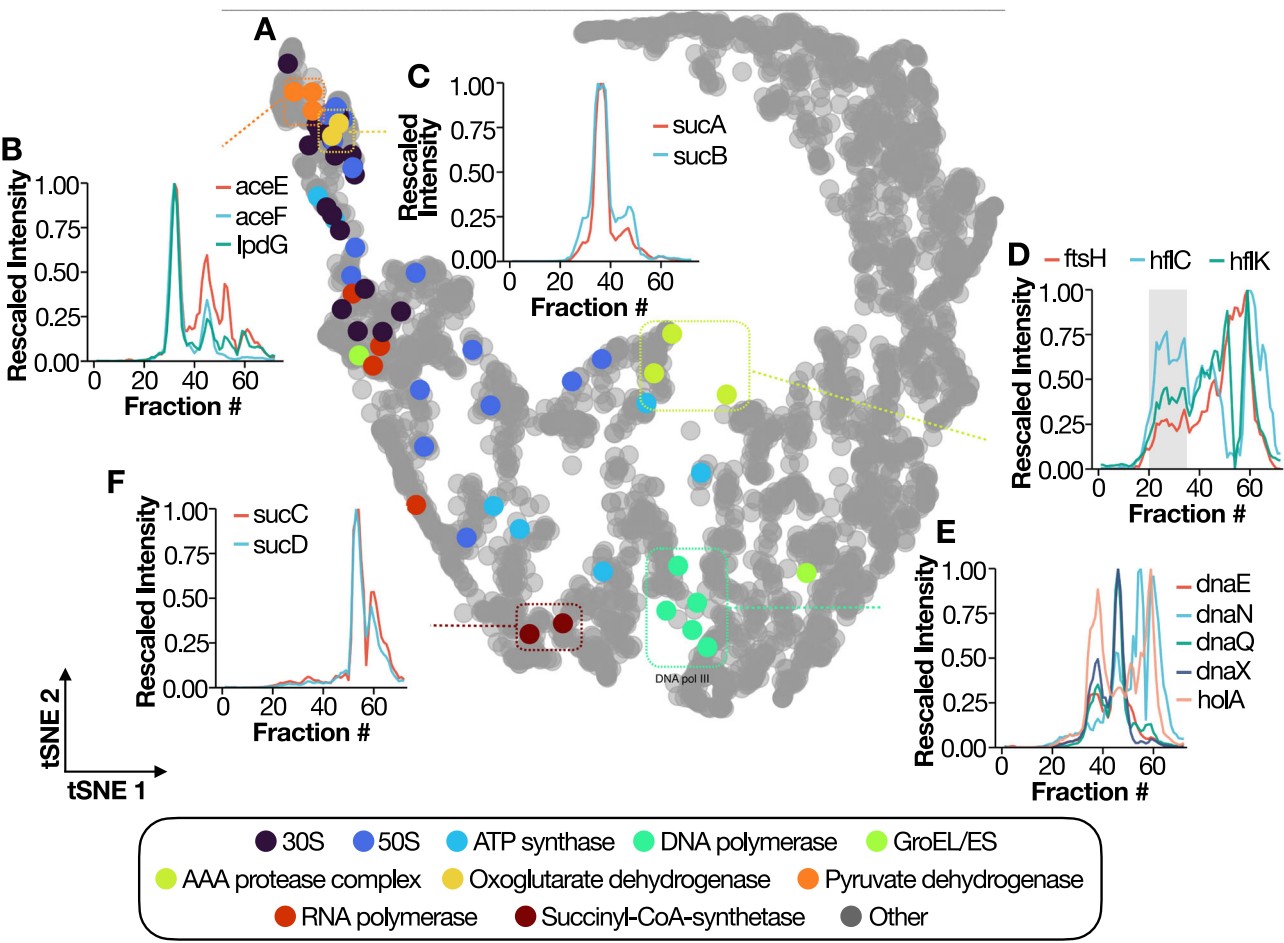

**Fig. 2 | *Pseudomonas* protein complexes identified in the SEC-MS data. A** t-SNE plot for all the *P. aeruginosa* proteins detected in the PAO1 control. Each dot represents an individual protein, while the color code represents membership in reported protein complexes. Representative coelutions are shown for the pyruvate dehydrogenase complex (**B**), Oxoglutarate dehydrogenase complex (**C**), AAA protease complex (**D**), DNA polymerase III (**E**), and succinyl-coA synthetase (**F**). The x-axis shows the fraction number, while the y-axis indicates the unit-rescaled intensity. The line color shows the various subunits.

depicted in Fig. 3E. For example, the biofilm formation pathway (KEGG id:pae02025) was enriched in both phage infected samples ($q \leq 0.01$). Several prior studies have highlighted the role of phages in regulating the formation of biofilms via modulation of polysaccharide production and perturbation of cell envelope biology[27,28]. We identified multiple proteins in this category having significantly decreased abundance in the high molecular weight region compared to their uninfected counterpart (Fig. 3F), suggesting a lower assembly state or complex reduction upon infection. Specifically, we observed >2-fold reduction in pslD, pslE, and pslG in the high-molecular-weight region. These proteins are members of a complex spanning from the inner membrane (pslG) to the outer membrane (pslD), which is required for the biosynthesis of exopolysaccharide[29]. Additionally, the uncharacterized proteins PA3346, PA2366, and PA1667, display a broad coelution profile across the molecular weight dimension in the control sample, which is typically associated with membrane proteins[12]. Notably, these peaks are largely depleted (>3-fold reduction) in the infected condition. Outer membrane proteins were particularly affected by Jumbophage infection with porins and multi-drug efflux proteins (KEGG pae02010: ABC transporters), displaying a significant reduction in interactions. For example, the MexAB–OprM complex, a key efflux pump[30], shows an almost complete reduction of the fully assembled complex (Fig. 3G). Importantly the MexAB-OprM complex was previously shown to be targeted by a Jumbophage closely related to *φ*KZ, called OMKO1[31], potentially representing a secondary receptor-binding site for *φ*KZ-like Jumbophages.

It is important to point out that changes we observed could either be beneficial for the phage to overcome its host, a host response to limit phage development, or simply be the result of pleiotropic regulators.

## Organization of *φ*KZ-like Jumbophage viral interactomes

The remodeling of host protein complexes can be the result of indirect rewiring of host cellular processes or direct interactions with phage proteins. Thus, we next investigated interactions directly involving phage proteins, including complexes containing both phage–host and phage–phage interactions.

Following SEC-MS and PPI prediction, we defined high-confidence interactions as those with a probability score of $\geq 0.75$. In total, we identified 292 interactions between pairs of *φ*KZ viral proteins and 6550 HPIs between *φ*KZ and PA01 proteins. *φ*PA3 showed a similar trend with 145 viral-viral and 3979 host–pathogen protein interactions (Fig. 4A). Topological analysis of these networks revealed a scale-free architecture (Fig. 4B), in line with previous reports that SEC-MS-derived networks present the same architectural features as networks derived from literature curated studies and large PPI databases[12,32,33]. It has been observed with smaller phages that genes within the same operon are often functionally related[34]. Accordingly, we evaluated the distribution of our predicted PPIs (by SEC-MS) in phage-infected PAO1 cells as a function of the genomic separation of their corresponding genes. Here, we find a wide variation in the genomic distance

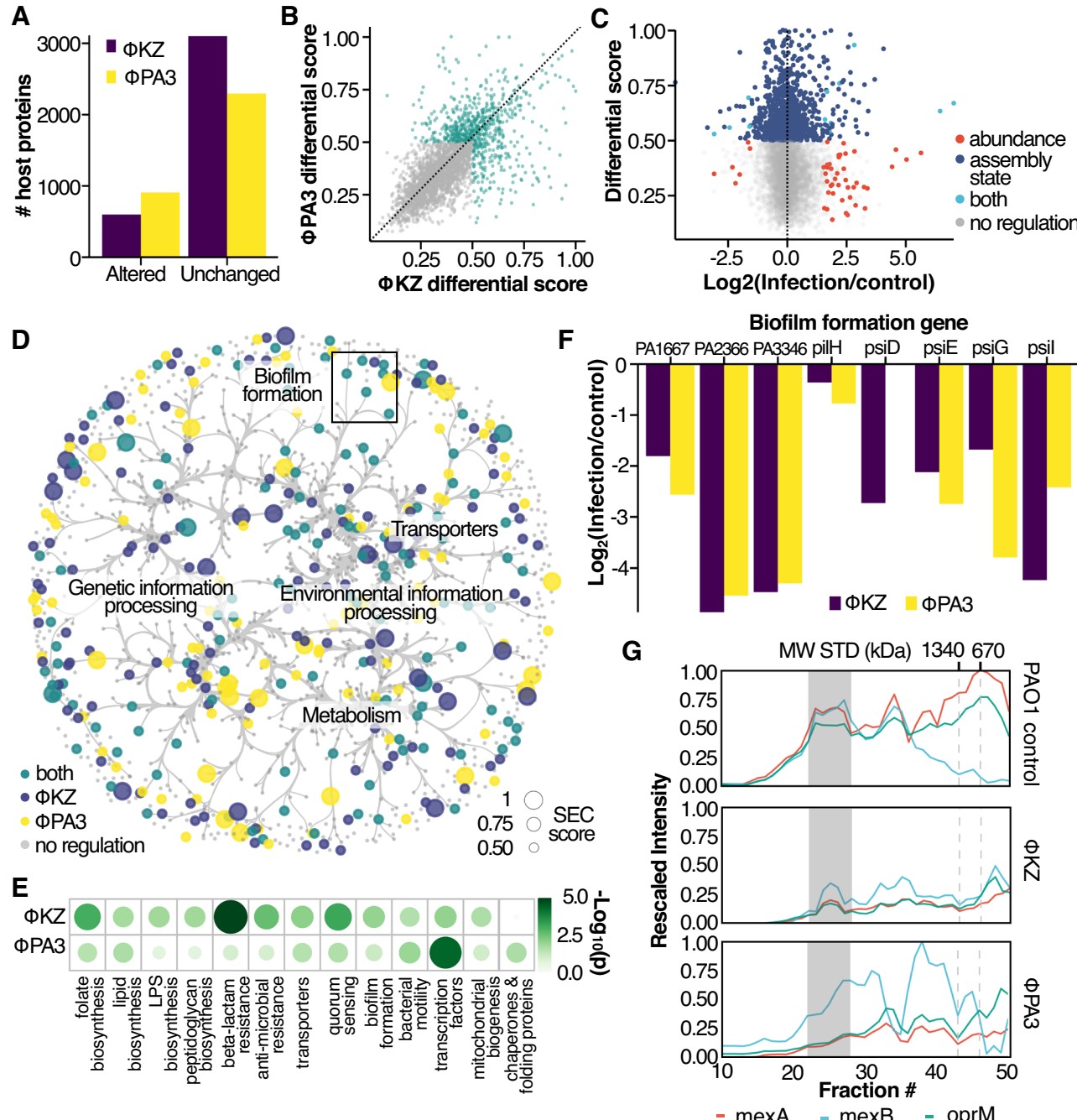

**Fig. 3 | Differential analysis of SEC-MS data. A** Altered host protein for each experiment. **B** Scatterplot of differentially regulated proteins. Axis represents the differential SEC score, while each dot represents a PAO1 protein. The color indicates significant regulation in either phage. **C** 2D plot of differential SEC score (*y*-axis) vs. LogFC from global proteome abundance (*x*-axis). Color represents the different regulation levels. Proteins highlighted in green are significantly regulated at the abundance level (Log2FC ≥ 2 and *q* ≤ 1%) and assembly state level (SEC score ≥ 0.5). Both phages are shown. **D** SEC-derived molecular network for PAO1 proteins. Node color represents the regulation status, while node size shows the SEC score (i.e., differential score). Edges are bundled using KDEEB. **E** Enriched KEGG terms for altered proteins. Node size and color represent the significance on a −log10 scale derived from a Fisher's exact test corrected for multiple testing using Benjamini–Hochberg. **F** Barplot representing the log2FC biofilm formation genes upon phage infection as compared to control. KZ-res m1 experiments for the assembled MW range (i.e., 2× monomeric weight for each protein). **G** Coelution plot for the efflux pump MexA/B-oprM. Different experiments are represented by the various subpanels.

between phage proteins that interact with other phage proteins (i.e., phage-phage interactions) (Fig. 4C), with some genes being separated by distances as large as 139 kb. For example, the two RNA polymerases in *ϕ*KZ are both composed of proteins expressed in different operons with a max distance of 112.761 kb (PHIKZ080–PHIKZ180 in the vRNAp). Thereby, our resulting PPI distribution confirms the general lack of synteny within the genomes of *ϕ*KZ-like Jumbophages and shows the SEC-MS approach is a particularly advantageous technique to query

phage-encoded protein complexes, agnostic to the overall genome organization (i.e., a guilt-by-association approach at the protein level).

## Data-driven identification of *ϕ*KZ-like Jumbophage protein complexes
The identified interactions allowed us to recapitulate several known complexes in the Jumbophage proteome, despite a limited number being described at present. For example, we recovered the non-virion-

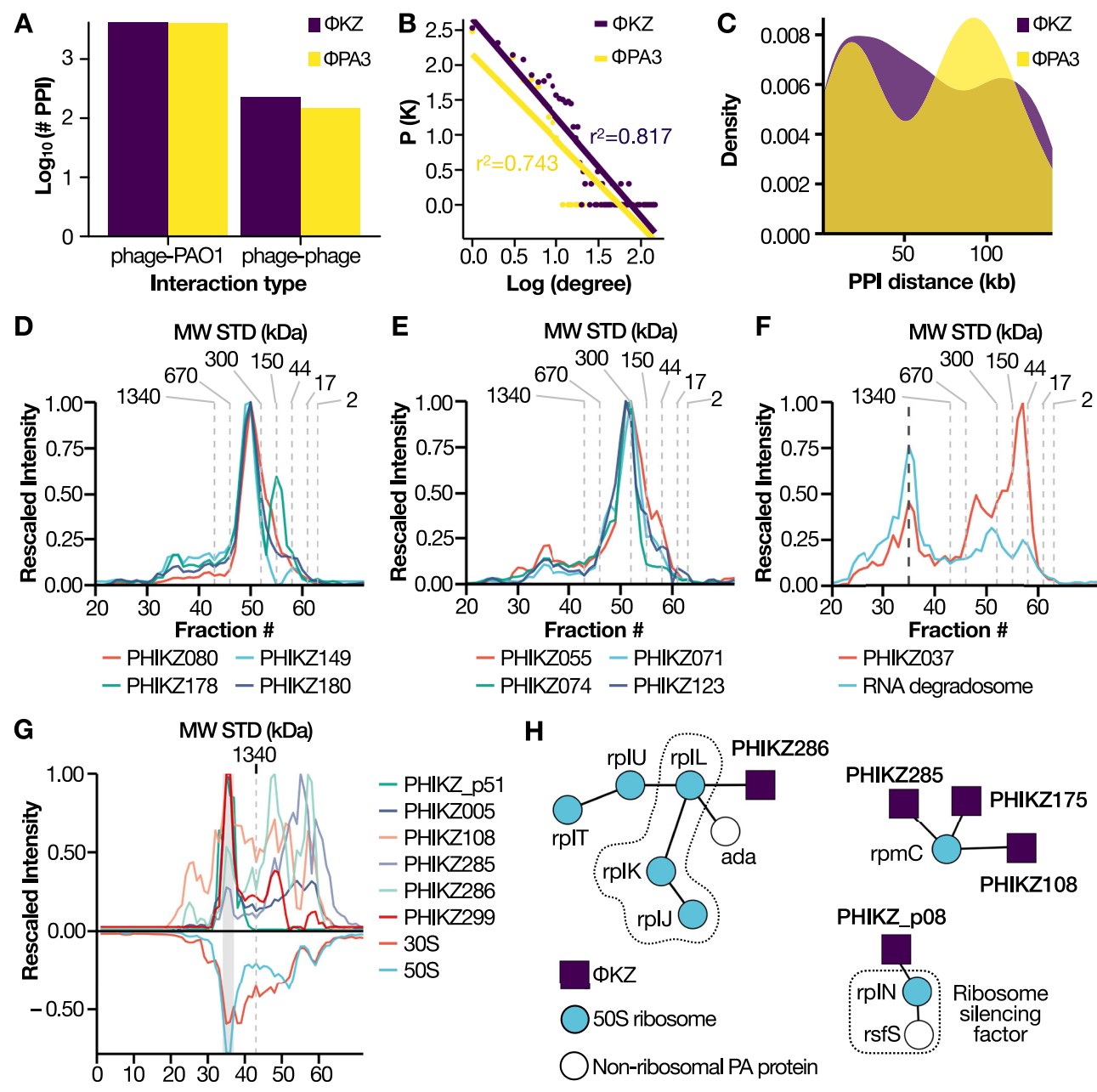

**Fig. 4 | Comparative analysis of φKZ and φPA3 interaction networks. A** Number of PPIs identified for each phage (*y*-axis), separated by interaction type. **B** Log degree distribution (*x*-axis) versus log frequency (*y*-axis). The line represents the best-fit power law with the correspondent *r²* value for each phage network. **C** Density plot illustrating the genomic distance in kb (*x*-axis) for the phage-phage interactions. Color codes represent the two phages. **D, E** Coelution plot for the φKZ non-virion-associated RNA-polymerase and the virion-associated RNA-polymerase. **F** Coelution plot for the RNA degradosome (cyan) and PHIKZ037 (red).

Degradosome intensity represents the average across all six identified subunits (rhlB, deaD, pnp, rhlE, hfq, rne). **G** Mirrorplot illustrating the coelution of φKZ proteins (upper panel) with the 70S ribosome (lower panel). **H** Interaction network for φKZ proteins and ribosomal subunits from the SEC XL-MS experiment. Edges represent an identified crosslink. The dashed line encloses known ribosomal structural components, node color represents whether a protein is a phage protein (blue), ribosomal component (purple), or a non-ribosomal *P. aeruginosa* protein (white).

associated RNA-polymerase[35] migrating at its expected molecular weight (apparent MW 271 kDa, correct MW ≈ 265 kDa) (Fig. 4D) as well as the virion-associated RNA polymerase[36] (apparent MW 300 kDa, correct MW ≈ 297 kDa) as shown in Fig. 4E. Previously described interactions between the phage and the host were also recovered by our approach, such as the interaction of PHIKZ037 with the RNA degradosome, which is involved in the accumulation of viral RNA[37] (Fig. 4F).

Building on the recovery of known phage protein complexes and the presence of several phage peak groups at the high-molecular-

weight (Supplementary Fig. S4), we sought to probe our dataset for previously undescribed shell-associated proteins. Only two proteins so far have been identified as fundamental for shell formation and function: the major shell protein PhuN[38,39] (gp54 in φKZ; gp53 in φPA3, gp105 in 201-φ2-1), which is the main building block of the shell complex[15,17,40], and the bipolar tubulin spindle protein phuZ which serves to stabilize the shell in the center of the bacterial cell[41]. Because the capsid docks on the shell prior to tail attachment and lysis[18], we are unable to differentiate phage proteins contained within the capsid from those that are ejected and associated with the shell by apparent

increases in molecular mass alone. To distinguish between these two cases, we performed two orthogonal control experiments using a cesium-purified virion sample and a shell-enriched sample (Supplementary Fig. S5A), which we used to filter the SEC-MS interactors to only proteins enriched in the shell sample and absent in the virion (Supplementary Fig. S5B, C). The six remaining proteins (PHIKZ_p22, PHIKZ036, PHIKZ111, PHIKZ_p64, PHIKZ232, and PHIKZ261) were then tested for association with the shell via fluorescence microscopy using PAO1 expressing mNeonGreen tagged constructs. Overexpression of most constructs resulted in diffuse localization outside of the phage shell (Supplementary Fig. S5D), with gp36-mNG and p64-mNG displaying filaments or puncta that were sometimes peripheral to the phage shell, but time-lapses revealed them to be mobile throughout the cell (see Supplementary Movies 1 and 2). Of note, these results do not conclusively exclude these proteins as potential shell components. Further confirmation of these results would be needed to address potential technical issues, such as disruption of protein localization by tagging or over-expression, shell association at an earlier or later stage of infection, or difficulty in detecting transient interactions. In addition, large (>MDa) and intact shell fragments have been shown to be mostly insoluble[39], likely resulting in the loss of many shell fragments prior to SEC and tightly bound shell-associated proteins as a result.

We then turned our attention to interactions between phage and host complexes. Interestingly, several $\phi$KZ proteins (PHIKZ005, PHIKZ108, PHIKZ285, PHIKZ286, PHIKZ299, and PHIKZ_p51) were predicted by our deep learning tool to be in complex with the fully assembled *P. aeruginosa* 70S ribosome (Fig. 4G). A recent preprint utilizing a low-resolution fractionation technique (Grad-seq) showed the presence of multiple proteins in the ribosome[42]. Notably, our co-fractionation experiments recovered most of them, albeit at a lower prediction confidence (0.5) than the one utilized to threshold the data (0.75).

To validate these $\phi$KZ proteins as ribosomal interactors, we performed cross-linking mass spectrometry (XL-MS)[43] on a pooled sample from the SEC fractions corresponding to the 70S ribosome (Supplementary Fig. S3). We identified 975 crosslinks in total (202 inter-links and 871 intra-links), covering several previously reported bacterial protein complexes (Supplementary Fig. S6). The XL-MS data recovered 24 *P. aeruginosa* ribosomal proteins (separated in 30S and 50S) of which 3 showed physical interaction with 5 $\phi$KZ proteins. Amongst these phage proteins, we recovered PHIKZ285, PHIKZ286, and PHIKZ108, which were predicted from the SEC-MS data to be in complex with the 70S ribosome. Moreover, we identified PHIKZ_p08 and PHIKZ175 as additional ribosomal interactors (Fig. 4H). PHIKZ286 bound the L1 ribosomal stalk (rplL, rplK, and rplJ), which has an important role in tRNA translocation[44] and is the contact site for several translation factors[45]. PHIKZ_p08 interacted with rplN bound to its ribosome silencing factor rsfS, which slows down or represses translation[46]. Finally, PHIKZ285, PHIKZ175, and PHIKZ108 were bound to rpmC, which is an accessory protein positioned near the exit site and required for triggering nascent polypeptide folding[47]. These findings demonstrate the power of SEC-MS to detect HPIs involved in critical aspects of host biology, however, further mechanistic characterization is needed to determine if such phage proteins manipulate host ribosomes or instead represent the active translation of the phage proteins.

## Identification of previously undescribed phage proteins by SEC-MS

The multiplexed nature of DIA allows unbiased sampling of the full precursor space[48], so we queried our data for the presence of peptides from previously undescribed phage proteins using a custom protein FASTA built with EMBOSS. We detected 4 previously undescribed proteins for $\phi$KZ (2 forward and 2 reverse ORFs) and 11 for $\phi$PA3 (8

forward and 3 reverse) (Fig. 5A, B). The authenticity of these previously undescribed proteins is supported by the detection of two or more proteotypic peptides for nearly all proteins (Fig. 5C) and reproducible detection of the same peptides in 15 or more consecutive fractions across independent experiments (Fig. 5D). All of these proteins showed reproducible quantitation between biological duplicate experiments ($n = 72$ per replicate), with an average protein-level correlation of 0.75 for $\phi$PA3 proteins and 0.82 for $\phi$KZ proteins (Fig. 5E). Most of these proteins migrated at a higher molecular weight than their predicted molecular weight, suggesting they may be associated with high-order assemblies (Fig. 5F). Some of the previously undescribed ORFs are further supported by a great degree of sequence overlap with homologs. The most staggering example is the $\phi$PA3 reverse sense ORF 56450–58417, which shows >70% sequence similarity with previously reported proteins from various *Pseudomonas* spp. phages ($\phi$KZ, Psa21, Phabio, 201$\phi$2-1, and PA1C) as shown in Supplementary Fig. S7A. Interestingly, all proteins showing ≥50% identity to 56450-58417 are previously reported or proposed phage RNA polymerase components (RNAP), such as PHIKZ074 (non-virion associated RNAp, UniprotID Q8SD88)[36,49,50]. To date, there is no experimental evidence of a nvRNAP in $\phi$PA3. To derive other putative members of this complex, we extracted the predicted interactors of ORF 56450–58417 (Fig. 5G) and performed BLASTp analysis to identify proteins showing homology to other Jumbophage RNA polymerase components. From this analysis, we selected four interactions, namely PHIPA3055, PHIPA3063, the previously undescribed ORFs 53811-55010, and 55491–56444. Recent work also detected these two ORFs, giving increased confidence in our findings[51]. Further manual curation of the genome file for $\phi$PA3 revealed an intron sequence between 53811–55004 and 55455–56444, leading to a single protein. Hence we renamed the 53811–55010 and 55491–56444 ORFs as 53811–56444.

The ORF 56450–58417 interacting proteins show >50% conservation with multiple Jumbophage proteins annotated as RNAP components (Supplementary Fig. S7B–D). Specifically, we identified homologs of both the $\beta\prime$ polymerase subunit (PHIPA3055 and ORF 56450–58417) as well as homologs of the $\beta$ subunit (ORF 53811–56444). The $\phi$PA3 protein PHIPA3063 displays 57% sequence similarity to PHIKZ068, an essential nvRNAp component that lacks structural similarity to known components of previously reported RNA polymerases[49]. Utilizing the position of the SEC peak, we estimated the nvRNAp MW in $\phi$PA3 to be ≈321 kDa (Fig. 5G). Assuming the lack of homodimers in the structure, the predicted MW for these four proteins was ≈283 kDa, suggesting a putative missing subunit. Of note, we did not identify 56450–58417 interactors corresponding to PHIKZ123, another $\beta$ subunit component, which could explain this observation. To explore the possibility of these proteins (PHIPA3055, PHIPA3063, ORF 53811–56444, and ORF 56450–58417) folding into an RNA polymerase-like assembly, we performed structural prediction of this peak group using AlphaFold2 multimer[52]. We aligned the best scoring model (ipTM + pTM = 0.86, Fig. S8) to the reported structures for the $\phi$KZ nvRNAP (PDB 7OGP https://www.rcsb.org/structure/7OGP and 7OGR https://www.rcsb.org/structure/7OGR)[50] as depicted in Fig. 5H. We reached a template modeling (TM) score of 0.72 using US-Align[53] and an average RMSD of 0.624 Å using MatchMaker[54] between our proposed $\phi$PA3 vRNAp and the $\phi$KZ RNAp (70GR), indicating a shared tertiary structure similarity between these two assemblies. As we obtained low distances for the $\beta$ and $\beta\prime$ subunits, we set to investigate the misaligned region at the C-term of the polymerase clamp (PHIPA3063 in $\phi$PA3 and PHIKZ068). Despite showing high sequence homology (68%), these two proteins share a large intrinsically disordered region (IDR) in the middle of the sequence (275-293 aa for PHIKZ068 and 277-301 aa PHIPA3063) as shown in Supplementary Fig. S9. The IDR likely enables flexibility in the central region, resulting in varied orientations for the folded C-term in PHIPA3063 following AlphaFold predictions.

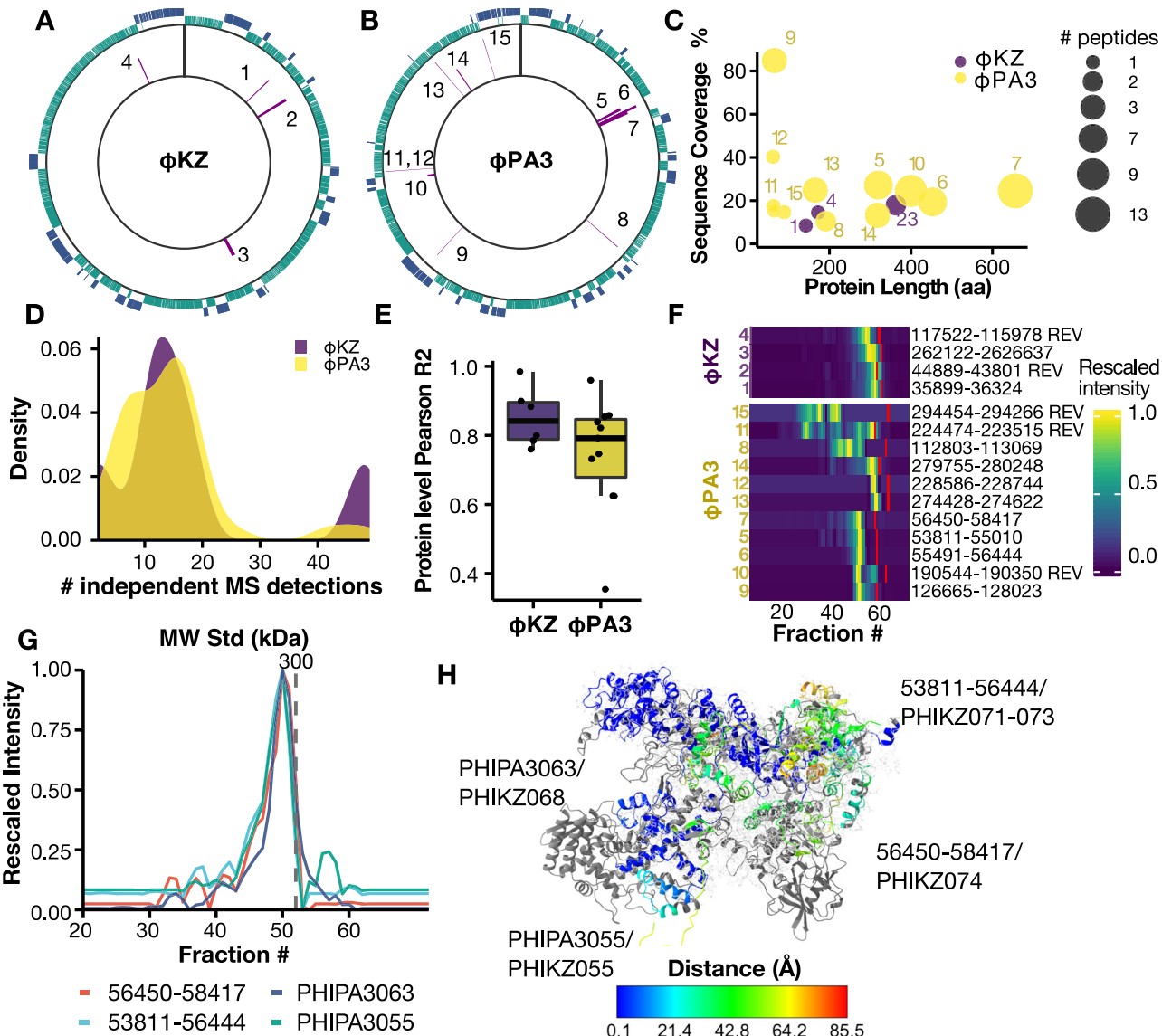

**Fig. 5 | Identification of previously undescribed phage proteins. A, B** CDS plot for φKZ and φPA3. Forward CDS are colored in green, while reverse CDS are represented in purple. Identified novel proteins are highlighted in the histogram (inner circle). **C** Scatterplot of protein length vs. percentage of sequence coverage in the SEC-MS experiments. The dot size represents the number of proteotypic peptides identified. **D** Distribution of the identifications (defined as a number of independent MS detections using a 1% peptide spectrum matching FDR) for the novel φKZ and φPA3 proteins. **E** Boxplot showing the Pearson correlation between the two replicates (*n* = 72). Each novel protein is represented as a dot. The box boundaries show the interquartile range (IQR) and its whiskers 1.5× IQR. Black lines represent the mean. **F** Heatmap representing the elution profile for all the novel ORFs. The *x*-axis represents the fraction number, while the cell color shows the unit-rescaled intensity. The solid red line represents the expected monomeric fraction. **G** Coelution profile for predicted nvRNAp in φPA3. **H** Superimposition of reported structure for the φKZ nvRNAp (stick, gray) and predicted structure for the φPA3 nvRNAp (ribbons). Chains are colored by their distance to the φKZ nvRNAp structure after superimposition.

## Discovery and validation of previously undescribed injected phage proteins

Although it is well-known that φKZ phages guard their genome from nucleolytic host-immune systems by building a proteinaceous shell[15], this structure is only visible after 20 min of infection. Little is known about how the phage genome is protected or packaged prior to shell assembly. To identify phage proteins proximal to the genome, with possible protective functions, we first determined a detailed virion proteome to allow distinction of virion proteins (injected) from newly synthesized proteins. We performed cesium chloride-gradient purification of φKZ coupled with deep peptide fractionation and long chromatographic acquisition (see Supplementary Methods for details). The 245 φKZ proteins identified in this dataset encompassed ≥90% of previously reported head proteins[55] (Supplementary Fig. S10A). To account for low-level contamination caused by cesium chloride-fractionation, we compared our enriched virion sample with the previously reported virion proteins to derive a ROC curve, which we used to select an intensity threshold maximizing recall of known virion proteins and minimize the false-positive rate (Fig. 6A). This filtering identified 81 significantly enriched proteins in total. This included 58/61 (95% recall) of those previously reported and added 23 proteins to the virion composition, which are strongly enriched over their corresponding protein abundance in a non-enriched sample (Fig. 6B). This drastic increase in protein number is dependent on the increased sensitivity sequencing speed of the MS utilized for acquisition as well as extensive offline sample fractionation prior to MS acquisition (see Supplementary Fig. S.10B).

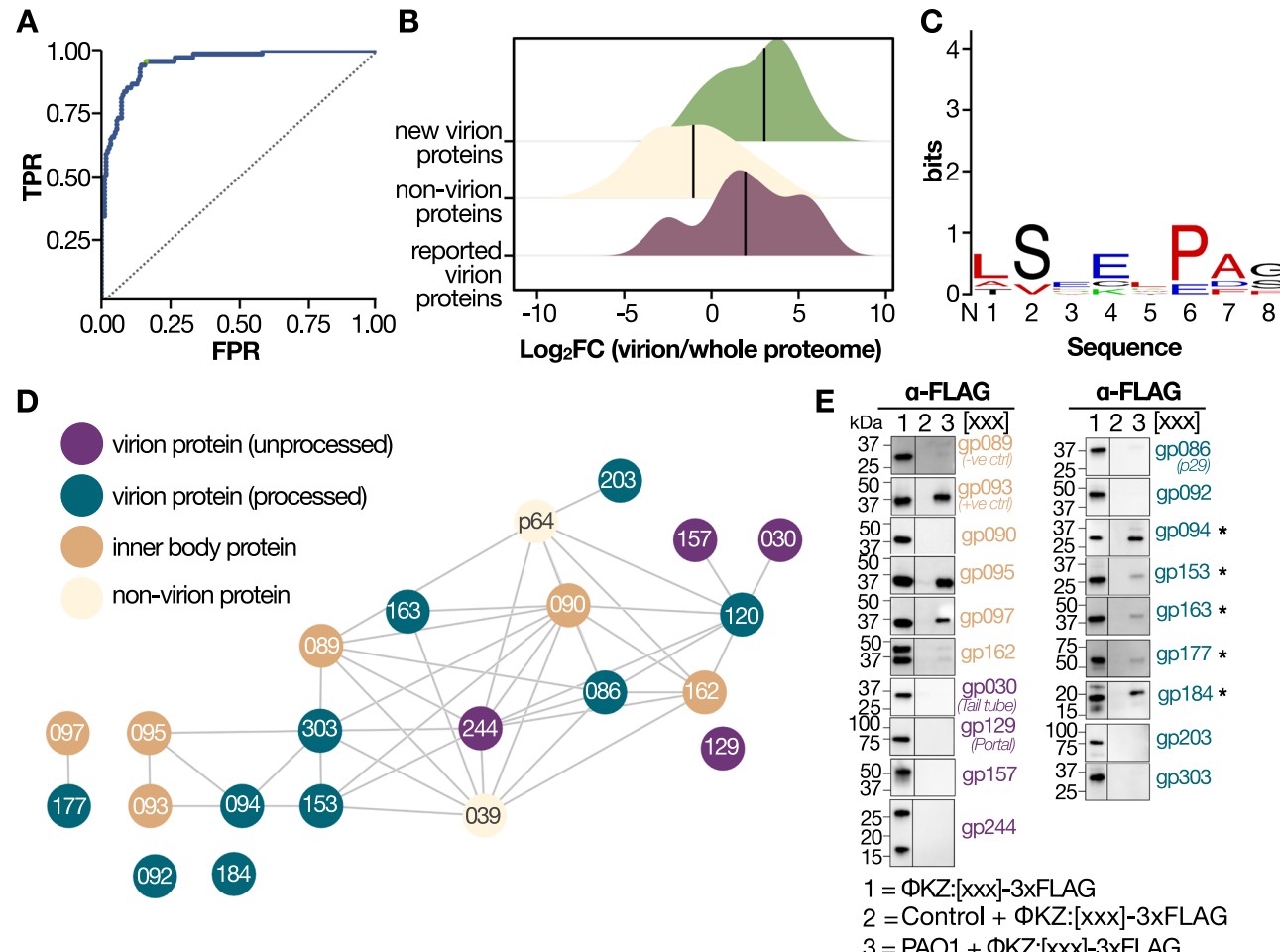

**Fig. 6 | Data-driven analysis of injected inner body proteins. A** ROC curve for virion MS (AUC ≈ 0.94) using as ground truth the prior reported virion proteins. Green highlights the selected threshold for maximum sensitivity at the lowest FPR. **B** Density plot representing the enrichment of virion proteins over a whole proteome infection experiment expressed as log2 fold change (*x*-axis). Different colors represent whether a protein was previously reported as a virion (purple), a novel from our virion dataset (green), or a non-virion (cream). **C** Sequence logo for the proteins in the IB interaction network. The *x*-axis shows the position from N to C term while the *y*-axis represents conservation in bits. **D** SEC-MS derived interaction network for the reported IB proteins (gp93/95/97). Color code represents the query protein (aquamarine), processed virion protein (dark green), non-processed virion proteins (dark purple), previously reported inner body proteins (orange) and proteins not identified in the virion MS experiment (light yellow). **E** Injection of phage proteins evaluated by western blot of 3× FLAG phage tagged proteins. gp089 (−ve ctrl) serves as a negative control, while gp093 (+ve ctrl) serves as a positive control (*n* = 1). Validated novel injected proteins are designated with an asterisk.

As prior work reported extensive gp175-driven proteolysis of the head and inner body (IB) proteins[55], we searched our purified virion data for evidence of unexpected termini (i.e., semi-tryptic peptides) with the goal of confirming prior reported cleavages and potentially identifying novel ones. We recovered 63 semi-tryptic peptides, of which 15 could be mapped to prior data[55] (≈ 40% overlap). Within our semi-triptic peptides, we identified 20 cleavages corresponding to the reported IB proteins (gp93/95/97) and 12 mapping to 9 unreported proteins. Of note, 8 cleavages could be mapped to gp94, gp177, and gp303 (see Supplementary data and Supplementary Fig. S10C). To identify consensus sequences within our set of IB interactions, we performed motif enrichment analysis using STREME[56]. We found that the LSxE consensus motif was enriched (BH-adjusted $p = 6e^{-5}$), corroborating the previously reported S/A/G-X-E motif while providing additional specificity in the P2 position (Fig. 6C).

Starting with only the virion proteome, we next queried the $\phi$KZ interaction network to identify putative injected proteins (Fig. 6D). Building on this data, we selected the interactors of the previously reported proteins (gp94, gp153, gp162, gp163, and gp177) for additional validation using our previously reported assay for evaluating injection[19]. In this assay, PAO1 cells expressing the protein of interest

with a FLAG tag are infected with wild-type $\phi$KZ, resulting in phage particles with labeled proteins. These phage particles are then used to infect gentamicin-treated PAO1 cells (i.e., cells where the translation is inhibited) allowing to evaluate injection using a western blot. By further lowering the interaction thresholds to positive predicted interactions (i.e., PPI probability ≥0.5 instead of 0.75 utilized to select high-confidence interactors), we further identified gp184 as an IB interactor and validated it as an injected protein. These experiments confirmed the injection of the previously reported IB proteins (gp93, gp95, gp97) and further validated the injection of most of their interactors (gp94, gp153, gp163, gp177, and gp184) as showcased in Fig. 6E. We note that among the 3xFLAG-tagged virion proteins tested in this study, the injection profile of inner body protein PHIKZ090 is inconsistent with a previous report[19]. In this prior study, it was reported that PHIKZ090 tagged at the C-terminus with mNeonGreen is injected into PAO1 cells upon infection by PHIKZ (as detected by fluorescence microscopy). We do not detect injection of PHIKZ090-3xFLAG via western blotting. The reasons for this discrepancy are unclear and may result from weaker expression of this construct (as compared to the controls PHIKZ093-3xFLAG and PHIKZ089-3xFLAG) and lower levels of packaging in the virion. As a consequence, the amount of PHIKZ090-3xFLAG

injected into PAO1 cells is reduced, likely to levels that are below the detection limit of the western blot assay. We acknowledge that poor expression of certain constructs is a limitation of this assay. Here, by using a highly sensitive MS of the virion combined with SEC-MS, we identify and validate the injection of eight proteins (three previously reported) that are highly abundant, found in the virion, and interact with the previously reported IB proteins. Overall, these proteins give us a starting point to unravel the interactome of the ejected phage genome and identify proteins that protect the genome from host nucleases.

## Discussion

Understanding the dynamics driving host and pathogen interactions and their dynamics upon infection is a crucial component to deepening our knowledge of the mechanisms regulating infection progression and outcome. To date, most proteomics studies of infectious diseases focused on the analysis of a few pathogen proteins by tag/antibody-based purification or the measurement of protein abundance variation in infected samples. Yet it is widely known that the pathogen proteome works as an ensemble through protein–protein interactions to hijack the host cell, which in turn regulates both expression and interaction between host proteins. Hence, a system-wide view of the intrinsic modularity of the pathogen proteome and how it quantitatively regulates host complexes is key to understanding pathogenic mechanisms at the molecular level.

In this study, we demonstrate the application of SEC-MS to systematically investigate pathogen proteome organization and host interactome plasticity upon Jumbophages infection of *P. aeruginosa*. $\phi$KZ-like phages (specifically $\phi$KZ and $\phi$PA3) are potent killers of *P. aeruginosa* (with a broad host range), making them timely alternatives to antibiotics with many $\phi$KZ-like phages already in clinical trials to treat bacterial infections. By obtaining an atlas of these phage interactomes, we can begin to construct a mechanistic understanding of the $\phi$KZ-like Jumbophage infections, ranging from viral composition to protein injection, transcription, and phage nucleus assembly and growth.

Our $\phi$KZ-like phage interactomes recapitulated prior evidence for the subdivision of Jumbophage proteomes into distinct assemblies, such as virion and non-virion-associated RNA polymerases, as well as the interaction with key host complexes such as the RNA degradosome. We expanded our knowledge on the phage interactions with essential host processes such as translation, where we identified phage proteins interacting with the ribosomal stem and ribosomal silencing factors. Moreover, while the lack of immediate genome organization hinders the prediction of functions for phage proteins, the deep coverage and unbiased nature of SEC-MS data offers a straightforward approach to identifying previously undescribed complexes and proposing putative functions. As an example, by using SEC-derived interactors of a de-novo predicted $\phi$PA3 protein (ORF 56450–58417), we identified a heterotetrameric assembly which is predicted to have strong structural homology to the reported nvRNAP in $\phi$KZ. This suggests that the unbiased nature of SEC-MS data allows for not only the discovery of an uncharacterized protein but also enables to probe of its putative function through the detection of new protein–protein interactions. Identifying such complexes will enable further investigation using structural and biochemical approaches. In addition to the identification of interactions, these maps offer the opportunity to further quantify host interactome remodeling and disentangle variation in expression from the assembly state. By comparing the *P. aeruginosa* interactome between infected and uninfected, we observed a large degree of changes during infection, with perturbation of similar complexes between the two Jumbophages suggesting conserved mechanisms of phage predation. While here we have a first draft of the KZ-like Jumbophage interactome, it is important to acknowledge the trade-off between specificity and throughput in interaction

identification in SEC-MS, which we mitigated by utilizing only high-confidence interactions for analysis. This step, while increasing the confidence in our PPIs identification, still does not allow for the complete removal of false-positive results due to the lack of a real ground-truth dataset, which is to be expected in large-scale fractionation experiments. Advances in deep learning models for prediction of interactions from co-fractionation mass spectrometry data and integration of orthogonal features (besides the coelution itself), such as predicted structure or function, are expected to improve prediction accuracy and reduce the false discovery rate for uncharacterized proteomes. Overall, the characterization of host-pathogen molecular networks remains challenging, but we provided the first interactome-wide study of infection progression using two models of $\phi$KZ-like phages in *P. aeruginosa*.

Wider application of SEC-MS is expected to significantly accelerate the characterization of pathogenic mechanisms by providing proteome-wide insights into the physical association between host and pathogen complexes, thus enabling the identification of novel druggable targets, host vulnerabilities, or guidance in the development of new biologicals.

## Methods
### Cloning
C-terminal 3xFLAG fusions of $\phi$KZ IB proteins pHERD30T plasmids encoding C-terminal 3xFLAG fusions of $\phi$KZ IB proteins (PHIKZ089, PHIKZ090, PHIKZ093, PHIKZ095, PHIKZ097, and PHIKZ162) were cloned as follows: the plasmids pHERD30T-IB-mNeonGreen (PHIKZ089, PHIKZ090, PHIKZ093, PHIKZ097, PHIKZ162) were digested with restriction enzymes XhoI and KpnI (upstream and downstream of the mNeonGreen sequence) to generate a pHERD30T-IB___ backbone (IB: PHIKZ089, PHIKZ090, PHIKZ093, PHIKZ097, PHIKZ162). An insert sequence corresponding to XhoI-GGGGS-3xFLAG-KpnI was digested with XhoI and KpnI to generate an insert fragment that was ligated individually with each corresponding backbone using T4 DNA ligase to generate the plasmids ((see: Supplementary Table 1 for a list of plasmid and Supplementary Table 2 for list of primers utilized). The pHERD30T plasmid encoding a C-terminal 3xFLAG fusion of PHIKZ095 was generated as follows—the plasmid pHERD30T -PHIKZ090-3xFLAG was digested with restriction enzymes SacI and XhoI to generate a pHERD30T___ -3xFLAG backbone. The plasmid pHERD30T-PHIKZ095-mNeonGreen (sequence: Supplementary Table 1) was digested with SacI and XhoI, and the smaller fragment corresponding to SacI-PHIKZ095-XhoI was extracted by DNA Gel extraction to obtain an insert fragment. The insert fragment was ligated with the pHERD30T___-3xFLAG backbone using T4 DNA ligase to create the plasmid pHERD30T-PHIKZ095-3xFLAG. All plasmid sequences were checked for correct assembly upstream and downstream of the insert sequence with Sanger Sequencing (Quintara Biosciences) using sequencing primers QB0068 (5′-ATGCCA-TAGCATTTTTATCC-3′) and QB0049 (5′-CCCAGTCACGACGTTG-TAAAACG-3′). C-terminal 3xFLAG fusions of $\phi$KZ virion proteins: pHERD30T plasmids encoding C-terminal 3xFLAG fusions of $\phi$KZ proteins (PHIKZ030, PHIKZ_p29, PHIKZ092, PHIKZ094, PHIKZ129, PHIKZ153, PHIKZ157, PHIKZ163, PHIKZ177, PHIKZ184, PHIKZ203, PHIKZ244, PHIKZ303) were generated as follows: the plasmid pHERD30T-mNeonGreen -3xFLAG was linearized via PCR using primers pHERD30T-mNG-3xF_F, pHERD30T-mNg-3xF_R to generate a pHERD30T___-3xFLAG backbone (removing the mNeonGreen sequence). The genes encoding $\phi$KZ proteins were amplified from purified $\phi$KZ particles (removing the stop codon) via PCR using the corresponding primers to generate insert fragments. These insert fragments were individually assembled with the pHERD30T___-3xFLAG backbone to generate the plasmids. All plasmid sequences were checked for correct assembly upstream and downstream of the insert sequence with Sanger Sequencing (Quintara Biosciences) using

sequencing primers QB0068 (5′-ATGCCATA GCATTTTTATCC-3′) and QB0046 (5′-TGTAAAACGACGGCCAGT-3′). Benchling files containing sequences of all constructs, attached primers, and sequencing files are reported in Supplementary Data Files.

## Plaque assay

Plaque assays were conducted at 30C with solid LB agar plates. Totally, 150 µL of overnight bacterial culture was mixed with 3 mL top agar (0.35% LB-Agar, 10 mM MgSO$_4$) and plated on bottom Agar (20 mL LB-Agar, 10 mM MgSO$_4$). Phage lysates were diluted 10-fold, and 2 µL spots were applied to the top agar after it had been poured and solidified.

## Bacterial culture

*P. aeruginosa* strains PAO1 were grown overnight in 3 mL LB at 37 °C with aeration at 175 rpm. Cells were diluted 1:100 from a saturated overnight culture into 100 mL LB with 10 mM MgSO$_4$ and grown for ≈2.5 h at 37 °C with aeration at 175 rpm. At OD600 nm = 0.5–0.6 (≈3e$^8$ CFU/mL), the cell cultures were infected with bacteriophage ($\phi$KZ or $\phi$PA3; MOI ≈ 1) on ice for 10 min (to allow complete adsorption of virions onto cells) and then incubated at 30 °C for 50 min (total time of infection 60 min). Thereafter, the cell cultures were transferred to pre-chilled 50 mL falcon tubes and centrifuged at 6000x*g*, 0 °C for 5 min. The supernatant was discarded, and cell pellets were washed twice with 5 mL ice-cold LB and combined.

After the final wash, the bacterial pellets were resuspended in a 5 mL ice-cold LB. The concentrated cell culture was flash-frozen in liquid nitrogen and subsequently mechanically lysed using a SPEX-freezer mill.

## Shell isolation via density centrifugation

The shell isolation was performed as we previously reported[39]. Briefly, we infected *P. aeruginosa* PA01 with $\phi$PA3 or $\phi$KZ for 60 min. The bacteria were mechanically lysed via Dounce homogenization in NP40 Lysis Buffer (50 mM Bis−Tris, 150 mM NaCl, 0.5% NP40, 5% glycerol, 5 mM DTT, 20 ng/µl Lysozyme, 1 mM EDTA, 1 mM EGTA−pH 6.5). The lysate was clarified at 16,000×*g* for 5 min, and the insoluble fraction was resuspended in wash buffer (20 mM Bis−Tris, 150 mM NaCl, 1 mM DTT, 1 mM EDTA, 2 mM MgCl$_2$−pH 6.5). This was subject to further 500×*g* (5 min) and 15,000×*g* (10 min) centrifugation with the insoluble fraction isolated and resuspended in wash buffer each time. The insoluble fraction of the last 15,000×*g* spin was retained as the final product. The shell-enriched sample was acetone precipitated using eight volumes of ice-cold acetone and incubated overnight. Following incubation, the protein pellet was washed thrice with ice-cold acetone and dried under a vacuum.

## Cesium gradient purification of phage virions

Bacteriophages ($\phi$KZ or $\phi$PA3) were propagated in LB at 37 °C with PAO1 as a host. Liquid growth curve experiments were used to ascertain the MOI of bacteriophage stock needed to ensure complete lysis of the bacteria following a substantial growth as ascertained by OD600 measurement. Growth curve experiments were carried out in a Synergy H1 micro-plate reader (BioTek, with Gen5 software). Cells were diluted 1:100 from a saturated overnight culture with 10 mM MgSO$_4$. Diluted culture (140 µl) was added together with 10 µl of 10× serial dilutions of bacteriophage stocks to wells in a 96-well plate. This plate was cultured with maximum double orbital rotation at 37 °C for 24 h with OD600 nm measurements every 5 min. Thereafter, the bacteriophage stock was added at the appropriate MOI to a 1:100 back-dilution of a saturated PAO1 overnight culture in 100 mL LB with 10 mM MgSO$_4$, and the bacterial culture incubated for 24 h (37 °C with aeration, 175 rpm). Totally, 5 mL of chloroform was added to the cultures in a fume-hood, and the cultures were incubated with chloroform for 15 min (37 °C, 175 rpm) to ensure maximum lysis of

bacterial cells. The cell cultures were transferred to 50 mL falcon tubes and centrifuged at 6000×*g* for 15 min to pellet bacterial debris. The supernatant (containing bacteriophages in high titer) was carefully transferred to a fresh set of 50 mL falcon tubes and centrifuged and 6000×*g* for 15 min to pellet any residual bacterial debris. The supernatant was transferred to fresh 50 mL falcon tubes with 2 mL chloroform. To obtain high-purity virion particles, a previously described protocol was followed[57]. The virions from the bacterial cell lysate were concentrated by slow stirring overnight at 4 °C in 1 M NaCl and 10% PEG (final concentration) and then pelleted (11'300×*g*, 4 °C, 30 min). Pellets were resuspended in 20 ml of SM buffer (50 mM Tris-HCl (pH 7.5), 100 mM NaCl, 8 mM MgSO$_4$, 0.002% gelatin) containing Complete Protease Inhibitor (Roche). The phage suspension (5.8 mL/tube) was layered onto CsCl step gradients composed of the following concentrations of CsCl: 1.59 g/ml (0.75 ml), 1.52 g/ml (0.75 ml), 1.41 g/ml (1.2 ml), 1.30 g/ml (1.5 ml) and 1.21 g/ml (1.8 ml). The buffer used throughout the gradient was 10 mM Tris-HCl (pH 7.5) and 1 mM MgCl$_2$. Tubes were spun at 31,000 rpm for 3 h at 10 °C in an SW41 rotor (Beckman Coulter ultracentrifuge), and the resulting phage band had a buoyant density of 1.36 g/ml. This fraction was collected and dialyzed against three changes of 50 mM Tris-HCl and 10 mM MgCl$_2$ at 4 °C. This ultra-purified phage stock was diluted in SM buffer, and its titer was assessed using plaque assays. Finally, the phage virion stock was acetone precipitated using eight volumes of ice-cold acetone.

## Bacterial infection and SEC sample preparation

Cryomilled samples were resuspended in ≈4 ml of SEC running buffer (50 mM ammonium bicarbonate and 150 mM NaCl pH 7.4) supplemented with protease inhibitors (Roche) and ultracentrifuged at 60,000×*g* for 30 min at 4 °C. The supernatant was concentrated to 100 µL using a 100 kDa molecular weight cutoff filter to simultaneously enrich for high-molecular-weight assemblies and deplete monomeric proteins. The concentrated sample was centrifuged once more at 10,000*g* at 4 °C to remove particles.

## Size-exclusion chromatography

Approx 1000 µg per sample (≈80–90 µL as estimated by Bradford's assay) were separated on an Agilent Infinity 1260 HPLC operating at 0.5 mL/min in SEC running buffer with a Phenomenex SRT-C1000 column connected and cooled at 4 °C. Seventy-two fractions of 125 µl were collected after 3.75 ml until 13 ml and the column was then washed with 2 column volumes (18 mL) of SEC buffer. The MW was estimated using a protein mixture (Phenomenex AL0-3042), while an *E. coli* 70s ribosome (NEB, cat nr P0763S) was used to estimate which fractions to use for ribosome XL-MS.

## SEC-MS proteomics sample preparation

The SEC samples were prepared as we previously reported[58] using a 96-well filter-aided sample preparation (FASP). The FASP filters were conditioned by washing twice with 100 µL of ddH$_2$O. SEC buffer was removed by centrifugation (1800×*g* 1 h), and proteins were resuspended in 50 µL of TUA buffer (TCEP 5 mM, Urea 8 M, 20 mM ammonium bicarbonate) and incubated on a thermos shaker (37 °C, 400 rpm) for 30 min. Cysteine residues were then alkylated by the addition of 20 µL CAA buffer (Chloroacetamide 35 mM, 20 mM ammonium bicarbonate) for 1 h at 25 °C in the dark. TCEP and CAA were removed by centrifugation (1800×*g*, 30 min), and filters were washed 3 times with 100 µL of 20 mM ammonium bicarbonate. Proteins were digested in 50 µL of 20 mM ammonium bicarbonate with 1 µg of tryspin per fraction. A 96-well receiver plate (Nucleon, Thermo-Fisher) was used to collect the peptides by centrifugation for 30 min at 1800×*g*. The filter plates were washed once with 100 µL of ddH$_2$O and centrifuged to dryness (1800×*g*, 60 min). The peptides from the receiver plate were transferred to protein LoBind tubes (Eppendorf),

and the corresponding well was washed with 50 μL of 50% acetonitrile (ACN) in ddH$_2$0 to increase the recovery of hydrophobic peptides. The combined resulting peptides per each fraction were vacuum dried and stored at −80 °C until MS acquisition. For each phage, 5 μL from each fraction were pooled together to generate a phage-specific library. Each sample-specific library was prepared on a C18 spin column (Nest). Following activation of the column with 1 column volume (CV) 100% ACN and wash with 2 CV of 0.1% formic acid, the peptides were bound to the column and eluted using a step-wise gradient of ACN from 5 to 25 (5% increases) in 0.1% triethylamine to account for the increased hydrophobicity of the XL peptides compared to not modified ones. A final fraction at 80% ACN was added to recover hydrophobic peptides.

### Proteomics sample preparation for virion-enriched protein pellets

Dried proteins were resuspended in 100 μL of 8M urea, 100 mM ammonium bicarbonate (ABC) pH 8.1. TCEP (Thermo Fisher) was added to 5 mM final concentration, and the samples were incubated at room temperature for 30 min. Reduced cysteines were alkylated with 10 mM chloroacetamide (CAA) for 30 min in the dark. Following alkylation, the urea was diluted to 1 M with 100 mM ABC and the proteins were digested with 2 μg of trypsin per sample for 14 h at 37 °C in a thermo-shaker (600 rpm). Digestion was stopped by acidification using 10% formic acid (FA), and the samples were desalted using a C18 spin column (Nest group). Briefly, columns were activated using 1 column volume (CV) of ACN and then equilibrated with 2 CV of 0.1% FA. Peptides were loaded twice and then washed with 3 CV of 0.1% FA. Elution was done using 0.5 CV of 50% ACN 0.1% FA and repeated twice. Samples were dried under vacuum and stored at −80 °C until acquisition.

### Crosslinking MS sample preparation

$\phi$KZ infection and SEC separation were performed as described above. Following separation, the SEC fractions corresponding to the 70S ribosome peak (F33–F38) were pooled. The was crosslinked for 1 h at RT using 5 mM DSSO from a freshly prepared 30 mM stock in water-free DMF. The reaction was quenched by the addition of ABC to 50 mM for 30 min at RT, and the proteins were precipitated using 8 volumes of ice-cold acetone. Following overnight incubation, pellets were washed 5 times with 8× volumes of ice-cold acetone and briefly dried under a vacuum. The pools were reconstituted in 8 M urea, 100 mM ABC and 5 mM TCEP and incubated for 30 min at RT. CAA was added to 10 mM final concentration, and the samples were incubated in the dark for 1 h. Urea was diluted to 1 M by the addition of 100 mM ABC, and the proteins were digested overnight with 2 μg of trypsin in a thermo shaker at 30 °C. Samples were acidified with 10% TFA, and high-ph tip fractionation was performed as we previously described[58]. Briefly, following activation, equilibration, and washing of the C18 resin, the elution was done using a step-wise gradient of ACN from 10 to 40 (5% increases) in 0.1% triethylamine to account for the increased hydrophobicity of the XL peptides compared to not modified ones. The resulting fractions were dried under a vacuum.

### SEC-MS and spectral library acquisition

Samples were resuspended in buffer A (0.1% FA), and approximately 200 ng were analyzed by DIA-PASEF on a Bruker TimsTOFpro interfaced with a Ultimate3000 UHPLC. For the SEC-MS experiment, the peptides were separated on a PepSep column (15 cm, 150 μm IID) using a 38-min gradient at 0.6 μl/min. Following loading, the peptides were eluted for 20 min with a 5% to 30% B (0.1% FA in ACN) in 20 min. The column was then washed for 5 min at 90% and high flow (1 μl/min) and re-equilibrated at 5% ACN for the next run. The peptides were sprayed through a 20 mm ZDV emitter kept at 1700 V and 200 μC. The mass spectrometer was operated in positive mode using DIA-PASEF acquisition[59]. Briefly, 4 PASEF scans (0.85 1/$K$0 to 1.30 1/$K$0) were acquired and divided each precursor range into 24 windows of 32 Da (500.7502−966.67502 $m/z$) overlapping 1 Da. Each of the fractionated samples (phage-specific libraries) was acquired in DDA-PASEF using a similar gradient composition except for the elution, which was performed in 90 min leading to a 120 min gradient. For DDA-PASEF, the ion mobility window and precursor range were matched to the DIA boundaries to allow for seamless library building and search.

### XL-MS data acquisition

The XL-MS samples were acquired on a Bruker TimsTOFpro interfaced with a Ultimate3000 UHPLC. The peptides were separated using a 118 minutes linear gradient. Following loading, the percentage of B (80% ACN in 0.1% FA) was increased from 2% to 8% in 5 min and then to 43% in 90 min. Residual peptides were eluted at 50% B for 10 min, and then the column was washed at 88% B for the remaining 13 min. The peptides were separated on a PepSep column (15 cm, 150 mm iid, 1.9 μm beads size). The mass spectrometer was operated in positive mode and data-dependent acquisition with the same source parameters as the SEC fractionated samples. To enrich for crosslinked peptides, a custom IM polygon was employed[60], and charge inclusion was enabled (3 + $to$8 + precursors). Precursors having nominal intensity above 20,000 were selected for fragmentation using an inverted collision energy of 23 eV at 0.73 1/$k$0 and 95 eV at 1.6 1/$k$0.

### SEC-MS data analysis

The DDA files were searched within the Fragpipe toolkit using MSfragger[61] v3.7 and the 'DIA-speclib-quant' workflow using the *Pseudomonas aeruginosa* pan proteome FASTA (5564 entries, proteome ID UP000002438, downloaded on the 05/22, https://www.uniprot.org/proteomes/UP000002438). For each phage, the correspondent FASTA nucleotide file was downloaded from GenBank (NC_004629.1 https://www.ncbi.nlm.nih.gov/nuccore/NC_004629 for $\phi$KZ and NC_028999.1 https://www.ncbi.nlm.nih.gov/nuccore/NC_028999 for $\phi$PA3), and EMBOSS was used for novel ORFs prediction (see 'Prediction of novel ORFs' section for details). The GenBank files were translated to protein level using BioPython and supplemented to the *Pseudomonas* FASTA. Carbamylation of cysteines was set as a fixed modification, while oxidation of methionine, N-term acetylation (peptide level), and pyro-glu formation were set as variable modifications. EasyPQP (https://github.com/grosenberger/easypqp, v 0.1.37) was used to generate a spectral library. Following phage-specific library generation, PAO1 precursors from all libraries were transferred to ensure the presence of the same PAO1 proteins with the same peptides across all DIA experiments using lowess for RT realignment. The DIA-PASEF data was searched with DIA-NN[62] v.1.7.1 using a library-centric approach. Identified spectrum with MS1 precursors within 10 ppm and MS2 precursors within 15 ppm were selected, and a second library was generated (double-pass mode). Quantification was set to robust (high accuracy), and cross-run normalization was disabled.

### XL-MS data analysis

XL-MS timsTOF files were converted to mgf using MSconvert v3.0.21072-998eff1c0 (developer build). MS1 peak picking was enabled, and the spectrum was denoised (top30 peaks in 100 m/z bins). Ion mobility scans were combined. Following the conversion, the peak files were searched in XiSearch[63] v1.7.6.7 using a fraction-specific FASTA containing only the protein ids identified by SEC-MS in the corresponding MW range. MS1 and MS2 tolerances were fixed to 10 and 15 ppm with 10 ppm of peptide tolerance. DSSO was selected as crosslinker (158.0037648 Da), and the correspondent oxidized, and amidated crosslinker was added as modifications. Link-FDR was fixed at 5% (boosted), and the resulting file were imported into XiView (https://xiview.org) for manual inspection of crosslinked spectrums.

## Data analysis for DDA-purified virion samples

TimsTOF DDA files were searched in MSfragger using the LFQ-MBR workflow. Cysteine carbamylation was selected as a fixed modification, while N-term acetylation and deamidation were enabled as a variable modification with a max of 3 variable modifications per peptide. Peptides of lengths 7–50 were searched again by a database of phage, *Pseudomonas aeruginosa*, plus contaminants. Decoys were generated by pseudo-inversion. Percolator was used for FDR-control at 1% PSM.

## Protein–protein interaction prediction from SEC-MS data

DIA-NN reports were filtered at 1% library *Q*-value, and to infer protein quantities, the top2 peptides yielding the highest intra-protein correlation were averaged (sibling peptide correlation strategy) across replicates for each condition. This step was performed across all samples to ensure the same peptides were used for every replicate and condition. The raw MS2 profiles were smoothed using a Savitzky–Golay filter and rescaled in a 0–1 range. A dot product matrix between all proteins was calculated, and proteins showing $r^2 \geq 0.3$ were selected as putative interactors for prediction. For every pair, we calculated 5 features: (i) sliding window ($q = 6$) correlation, (ii) fraction-wide intensity difference, (iii) peak shift, (iv) Euclidean distance, and (v) contrast angle dot-product.

For prediction, we utilized a fully-connected neural network implemented in Tensorflow v2.12.0 (https://www.tensorflow.org). Briefly, we set the input layer as a number of features (147), followed by a fully connected layer with 100 neurons and a dropout layer (0.2%), and a fully connected layer with 72 neurons. A final output layer using sigmoid as an activation function was used for classifying co-eluting and not-coeluting proteins. For training, a previously reported dataset was used[32]. To select for positive, we utilized protein pairs in STRING using a combined score of 0.9 and experimental evidence, while negative were randomly selected. The DNN model was trained for 100 epochs using ADAM (learning rate = 0.001) and binary cross-entropy as a loss function. Early stopping (patience = 20) was utilized to avoid overfitting. To further removed spuriously co-eluting PPIs after the prediction step, we calculated an equal number of decoy PPIs by randomly sampling the remaining proteins and utilizing the DNN model to predict their coelution probability. We then utilized these two distributions to perform target-decoy competition using posterior probabilities as we previously described[21] at 5% FDR, and the final interaction table was further filtered using a combined interaction probability of ≥0.75.

## ORFs prediction from nucleotide FASTA

EMBOSS v6.6.0.0 subroutine getorf was used to predict open reading frames (ORFs) with a minimum size of 50 AA. Existing annotated genes were removed from the predicted ORFs using bedtools subroutine subtract, allowing us to differentiate between existing and novel ORFs.

## Structural prediction and alignment for φPA3 vRNAp

Protein complex prediction was performed using AlphaFold 2 (https://github.com/deepmind/alphafold). AF2 was run with full database size and the multimer preset. OpenMM energy minimization was performed to generate relaxed models, and 5 models per complex were generated. Models were ranked by ipTM + TM, and the PAE and LDDT were extracted for visualization. Each complex was submitted as a FASTA file, with proteins ordered from the longest to the shortest sequence. The alignment was performed using US-Align[53] (https://zhanggroup.org/US-align/), and the oligomer option was selected. Alignments of predicted complex structures (φKZ vRNAp and 4 proteins φPA3 vRNAp) were performed by multiple structure alignment (MSTA) using US-align with default parameters, and a TM-cutoff of 0.45 was used to estimate topological similarities between the two structures. For visualization purposes, the structure of vRNAp (70GR)

without PHIKZ123, which lacked homolog identification in φPA3, was used as a template in MatchMaker.

## Fluorescent microscopy of putative shell components

0.8% agar pads were supplemented with 0.5 µg/mL DAPI for phage DNA staining *P. aeruginosa* strain PAO1 expressing each of the fluorescent shell candidate constructs was grown in liquid culture supplemented with up to 0.05% arabinose (depending on optimal conditions for each construct) to induce construct expression until an OD of 0.5, and subsequently infected with φKZ lysate for 50 min at 30 °C before imaging. Microscopy was performed on an inverted epifluorescence (Ti2-E, Nikon, Tokyo, Japan) equipped with a Photometrics Prime 95B 25-mm camera and the Perfect Focus System (PFS). Images were acquired using Nikon Elements AR software (version 5.02.00). Cells were imaged through channels of phase contrast (200 ms exposure, for cell recognition), blue (DAPI, 50 ms exposure, for phage DNA), and green (GFP, 200 ms exposure, for mNeonGreen constructs) at 100× objective magnification. Final figure images were prepared in Fiji (version 2.1.0/1.53c)[64].

## Generation of φKZ particles packaged with 3xFLAG fusions of φKZ virion proteins

φKZ particles packaged with virion proteins bearing a C-terminal 3xFLAG-tag were generated by adapting a protocol used to generate φKZ particles packaged with mNeonGreen-tagged inner body proteins[19,65]. PAO1 cells transformed with the appropriate pHERD30T – (PHIKZxxx) – 3xFLAG construct were grown overnight in 3 mL LB supplemented with gentamicin (50 µg/ml) at 37 °C with aeration at 175 rpm. Cells were diluted 1:100 from a saturated overnight culture into 5 mL LB supplemented with MgSO₄ (10 mM) and Gentamicin (50 µg/ml) and grown for ≈2.5 h at 37 °C with aeration at 175 rpm. At OD600 nm = 0.5–0.6 (3E8 CFU/mL), the bacterial cultures were infected with φKZ (WT, MOI ≈ 1) for 2.5 h. Thereafter 1 mL of chloroform was added to the cultures in a fume-hood, and the cultures were incubated with chloroform for 15 min (37 °C, 175 rpm). The cell cultures were transferred to 15 mL falcon tubes and centrifuged at 6000×g for 15 min to pellet bacterial debris. The supernatant (containing bacteriophages in high titer) was carefully transferred to a fresh set of 50 mL falcon tubes and centrifuged and 6000×g for 15 min to pellet any residual bacterial debris. Thereafter, 4 mL of the supernatant was filtered and concentrated (≈10×) using Amicon-100 centrifugal filters to remove excess 3xFLAG-tagged proteins. The concentrated supernatant was used for western blot experiments.

## Western blot and blot analysis

PA01 cells were grown overnight in 3 mL LB at 37 °C with aeration at 175 rpm. Cells were diluted 1:100 from a saturated overnight culture into 5 mL LB with 10 mM MgSO₄ and grown for 2.5 h at 37 °C with aeration at 175 rpm. Upon reaching 0.5 OD (600 nm), gentamicin was added (50 µg/ml), and the cells were chilled on ice for 5 min to stall translation. described, and upon reaching 0.5 OD (600 nm), gentamicin was added (50 µg/ml), and the cells were chilled on ice for 5 min to stall translation. Thereafter PAO1 cells (≈1 OD equivalent) were infected with φKZ particles packaged with virion proteins bearing a C-terminal 3xFLAG-tag (MOI ≈ 1) on ice for 10 min (to allow complete adsorption of virions onto cells) and then incubated at 30 °C for 15 min. Thereafter, the cell cultures were transferred to pre-chilled 15 mL falcon tubes and centrifuged at 6000×g, 0 °C for 5 min. The supernatant was discarded, and the cell pellet was washed twice with 2 mL of pre-chilled (0 °C) LB to remove excess unbound virions. The cell pellet was lysed in 100 µL of lysis buffer (20 mM Tris, pH 7.5, 150 mM NaCl, 2% glycerol, 1% TTX-100, CompleteMini EDTA-free protease inhibitor cocktail). The lysed suspension was further sonicated on ice using a Q125 sonicator (10 pulses, 1 s ON, 1 s OFF, 30% amplitude). The cell lysate was centrifuged at 15,000×g (15 min, 0 °C) to remove cellular debris. The

clarified cellular lysate (100 μL) was boiled with 33 μLL of 4× Laemmli Buffer (with Beta-mercaptoethanol) for 10 min. 14 μLL of lysate samples were loaded. For virion control samples, 10 μLL of purified virions were boiled with 3.3 μLL of 4× Laemmli Buffer (with Beta-mercaptoethanol) for 10 min, and 2 μLL of samples were loaded. SDS-PAGE gels were run with running buffer (100 mL 10× Tris-Glycine SDS Buffer, 900 mL Milli-Q water) at 130 V for 1 h (constant voltage setting). The SDS-PAGE gels were transferred onto 0.2 μM PVDF membranes using a wet transfer (Transfer Buffer: 100 mL 10× Tris-Glycine Buffer, 200 mL methanol, 700 mL Milli-Q water; 100 V, 1 hour, 4 °C). The membranes were incubated with blocking buffer (5% Omniblock milk, non-fat-dry in 1× TBST (200 mL Tris Buffer Saline, 0.20 mL Tween-20)) for 1 h at room temperature. Thereafter the blocking buffer was discarded, and the membranes were incubated with 1:1000 dilutions of mouse anti-FLAG M2 antibody (Sigma-Aldrich) in 1× TBST (overnight, 4 °C, with constant shaking). Thereafter the membranes were washed thrice for 10 min with TBST and incubated with 1:3000 dilution of Goat anti-mouse HRP (Ref: 62-6520; Lot: XD347166 Invitrogen) in blocking buffer for 1 h at room temperature with constant shaking. Finally, the membranes were washed thrice for 10 min with TBST and incubated with Clarity Western ECL substrate. Membranes were imaged on an Azure 500 imager.

### Statistics and reproducibility

No statistical method was used to predetermine sample size, no single fraction or replicate was excluded from the final analysis, and each SEC-MS was not randomized to avoid MS signal carry-over and increase reproducibility.

### Reporting summary

Further information on research design is available in the Nature Portfolio Reporting Summary linked to this article.

## Data availability

The supporting MS data is available via Massive with the identifier MSV000091715. The PhageMap database is freely accessible at https://phagemap.ucsf.edu/. The Alphafold2 predicted structures are available on GitHub at https://github.com/anfoss/Phage_data. The source data is included in this publication. Source data are provided with this paper.

## Code availability

The utilized software for prediction of PPIs from SEC data is freely available at https://github.com/anfoss/PPIprophet/ and at Zenodo at https://doi.org/10.5281/zenodo.8161692.

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

## Acknowledgements

This work was supported by an NIH grant 1R01AI167412 (J.B.D. and D.L.S.) and 1R01AI171041 (J.B.D and D.A.A.). D.M. is supported by the NIH Ruth L. Kirschstein National Research Service Award 1F32GM149125-01. We thank Prof. James Wells at UCSF for the usage of the HPLC used to perform the size-exclusion experiments. We thank Natalie Whitis for assistance with operating the SPEX-freezer cryo-mill. Molecular graphics were performed with UCSF ChimeraX, developed by the Resource for Biocomputing, Visualization, and Informatics at the University of California, San Francisco, with support from the National Institutes of Health R01-GM129325 and the Office of Cyber Infrastructure and Computational Biology, National Institute of Allergy and Infectious Diseases. Figure 1A, B was prepared with Biorender.

## Author contributions

A.F.: Performed proteomics sample preparation and analysis of all MS data, developed PhageMAP, and wrote the paper. D.L.S., J.B.D., D.A., N.K.: Conceptualization, supervision, writing, and funding acquisition. A.P.: Novel ORF prediction. D.M., C.K., B.G.: Phage infection experiments, virion enrichment, microscopy, and W.B. for injected proteins. E.N. and M.M.: Performed shell enrichment. Y.L.: Critical input in revising the paper. All co-authors contributed to reviewing and editing the paper.

## Competing interests

The Krogan Laboratory has received research support from Vir Biotechnology, F. Hoffmann-La Roche, and Rezo Therapeutics. Nevan Krogan has previously held financially compensated consulting agreements with the Icahn School of Medicine at Mount Sinai, New York,

and Twist Bioscience Corp. He currently has financially compensated consulting agreements with Maze Therapeutics, Interline Therapeutics, Rezo Therapeutics, and GEn1E Lifesciences, Inc. He is on the Board of Directors of Rezo Therapeutics and is a shareholder in Tenaya Therapeutics, Maze Therapeutics, Rezo Therapeutics, and Interline Therapeutics. D.L.S. has financially compensated consulting agreements with Maze Therapeutics and Rezo Therapeutics. The other authors declare no competing interests
