## [Peer Review File · Nature Communications]

Next-generation proteomics for quantitative Jumbophage-bacteria interaction mappingREVIEWER COMMENTS

Reviewer #1 (Remarks to the Author):

Fossati et al describe a strategy for mapping host-pathogen protein-protein interactions at scale using co-fractionation and mass spectrometry. Their system is applied to infection of *Pseudomonas* by jumbophage where they discover large numbers of host-pathogen and pathogen-pathogen interactions, of which a number involving the ribosome are specifically investigated by cross-linking. They go on to characterize virion packaged and injected proteins.

Overall, my view is the study is very strong. The manuscript is very and well illustrated, and the technical aspects look to be of a very high quality. I think the crosslinking data for the ribosome is particularly convincing. They provide a nice web interface to browse the results which is welcome. I think that the approach taken will be broadly applicable in other host-pathogen systems and should therefore be of general interest. As such, I think this report could be published in fairly short order. The comments below are generally some suggestions that might help to improve clarity here and there but they are squarely in the minor category. Note, that my comments focus mainly on the interaction and MS component as I know little about phage infection biology.

1. In the introduction they state that exogenous expression of pathogen proteins limits AP-MS based host-pathogen approaches. This is true to a large extent although some systems have been published (albeit less frequently) where the pathogen proteins are tagged endogenously in the pathogen making this comment slightly overwrought in my view. The HIV interaction paper of Brian Chait & Mark Muesing (Nat Micro 2016) comes to mind but I think there are others. This does not limit the motivation as their other points stand, and this endogenous tagging is rather difficult.

2. To assess changes to interactions in the data on infection they use a previously published approach (SEC differential score PCprophet). While I understand this is prior work I think it would be helpful to the reader to add few sentences describing the basis for this score (as much that follows stands on this understanding and there are many ways to do this).

3. In fig 3c they compare differential score from the SEC data with protein abundance changes. I am wondering where this protein abundance data came from. Was this measured separately from the SEC? or was this using the SEC data by summing over fractions? I don't see any specifics about this in results or methods. Please clarify.

4. They discuss the scale-free topology of their interaction set but the figure presented (4B) seems atypical for this claim. Could the authors plot log-log with including a power law fit (i.e. similar to the Huttlin paper they cite in this section)?

5. For fig 5F the plot the elution fractions of novel orfs to support the claim that they run higher than expected MW and are therefore in complex. But the monomer molecular weights are not shown on the plot so I think it cannot be interpreted in this way. Please add the monomer molecular weights (or expected fraction or whatever)

6. Proteomexchange access was not available for review

signed

Ben Collins

Reviewer #2 (Remarks to the Author):

The manuscript "Next-generation interaction proteomics for quantitative Jumbophage-bacteria interaction mapping" is a very interesting piece of work and very much-needed research in the field. The work is quite impressive and very much accessible.

The proteome work is very important to detect and determine the function unit of the phage and

its host. The use of SEC-MS to study the proteome of the phage during their infection is very informative and valid study. Thus, the paper can be accepted in its native form. No further comments are recommended.

Reviewer #3 (Remarks to the Author):

The manuscript from Fossati et al describes application of a new mass spectrometry method to analyse viral-host interactions, using *Pseudomonas aeruginosa* and two jumbophages, phiKZ and phiPA3.

There is an abundance of new data that appears thoroughly analysed. Known complexes are found as expected, which builds confidence in the approach, and new predictions are validated. Overall this should be of interest to those working in proteomics, phages, and viral-host interactions.

As a non-expert in MS, I will say that I found the manuscript rather opaque. The final impression is that thorough analysis has been performed, but is largely inaccessible to the generalist reader.

Specific comments:

Line 135 - can the authors tell us how similar the two phages are? What is the shared identity, are they taxonomically distinct?

Line 166 - please briefly explain why 60 min was chosen

Line 177 - why not do an uninfected control, or phage lysate control? Can you please state whether the KZ-resistant strain is also PA3 resistant?

Line 263 - "majority of these single-peak proteins" can you please provide numbers.

Line 320 - what does a t-SNE plot show, is the shape significant in some way?

Figure 3 - I think I am correct in saying these data are comparing the phage+bug experiment vs the phage+resistant bug experiment. How are the two biological replicates handled? Are they represented here? How do the replicates compare?

Line 606 - I keep wondering about the use of "interactions". The experiment simply analyses the protein content of fractions from a SEC experiment. The proteins in the same fraction cannot be said to interact, they are simply co-eluting, and could be doing so away from expected molecular weight due to the abundance of stickiness, or co-factors, in those fractions, rather than a specific interaction. Doesn't there need to be adequate and cautious consideration of the truth of these interactions. Perhaps some cunning stats can be employed to provide confidence (and indeed perhaps has been, but escaped this reviewer).

Line 609 - following on from above, these numbers are surprising. "292 interactions between pairs of KZ viral proteins and 6550 host-pathogen between KZ and PA01 proteins". Is this saying 292 interactions between phage proteins (believable) and 6550 interactions phage+host? Isn't 6550 astonishingly high? You say earlier (line 240) you detected 280 KZ proteins. $6550/280 \sim 23.4$. Do these data say each phage protein on average interacts with 23.4 host proteins? That doesn't seem reasonable. Is there precedent for this large an interaction network, or is this simply co-elution without true biological interaction? Can these numbers be justified as reasonable, to bat away doubt that a lot of the "interactions" are actually just noise?

Line 737 - Really good to see separate validation of interactions with the ribosome. Nothing negative here - I think it's a lovely experiment.

Typos lines 109, 165, 704, 733, 734, 769

REVIEWER COMMENTS

Reviewer #1 (Remarks to the Author):

Fossati et al describe a strategy for mapping host-pathogen protein-protein interactions at scale using co-fractionation and mass spectrometry. Their system is applied to infection of *Pseudomonas* by jumbophage where they discover large numbers of host-pathogen and pathogen-pathogen interactions, of which a number involving the ribosome are specifically investigated by cross-linking. They go on to characterize virion packaged and injected proteins.

Overall, my view is the study is very strong. The manuscript is very and well illustrated, and the technical aspects look to be of a very high quality. I think the crosslinking data for the ribosome is particularly convincing. They provide a nice web interface to browse the results which is welcome. I think that the approach taken will be broadly applicable in other host-pathogen systems and should therefore be of general interest. As such, I think this report could be published in fairly short order. The comments below are generally some suggestions that might help to improve clarity here and there but they are squarely in the minor category. Note, that my comments focus mainly on the interaction and MS component as I know little about phage infection biology.

1. In the introduction they state that exogenous expression of pathogen proteins limits AP-MS based host-pathogen approaches. This is true to a large extent although some systems have been published (albeit less frequently) where the pathogen proteins are tagged endogenously in the pathogen making this comment slightly overwrought in my view. The HIV interaction paper of Brian Chait & Mark Muesing (Nat Micro 2016) comes to mind but I think there are others. This does not limit the motivation as their other points stand, and this endogenous tagging is rather difficult.

We added references for the aforementioned paper and acknowledge that AP-MS using endogenous tagging has been done previously done for small pathogens with the sentence (line 101-106)

'While some of these limitations have been partially overcome by the introduction of endogenous tagging within the viral genome, this approach has been mostly limited to small viruses. Both endogenous tagging and ectopic expression are labor-intensive processes that require the generation of numerous plasmids and hundreds or thousands of individual purifications to comprehensively probe protein-protein interactions for an entire viral proteome.'

2. To assess changes to interactions in the data on infection they use a previously published approach (SEC differential score PCprophet). While I understand this is prior work I think it would be helpful to the reader to add few sentences describing the basis for this score (as much that follows stands on this understanding and there are many ways to do this).

We added the following sentence to provide a high-level description of the differential scoring employed (line 527-530)

'Differential analysis of two SEC profiles using PCprophet provides the SEC differential score which represents the variation of complex intensity (stoichiometry) or peak position (assembly state).'

3. In fig 3c they compare differential score from the SEC data with protein abundance changes. I am wondering where this protein abundance data came from. Was this measured separately from the SEC? or was this using the SEC data by summing over fractions? I don't see any specifics about this in results or methods. Please clarify.

The abundance data is derived from a single shot DIA-PASEF analysis of the same cell lysate utilized for SEC fractionation. To clarify this point we have added the following text to the section describing this figure panel (line 542-544)

'When compared to an independent whole cell lysate-proteome protein abundance measurement of the same cell lysate'

4. They discuss the scale-free topology of their interaction set but the figure presented (4B) seems atypical for this claim. Could the authors plot log-log with including a power law fit (i.e. similar to the Huttlin paper they cite in this section)?

We acknowledge that this was an oversight on our end, and fixed the plot accordingly. We enclose it here too to facilitate revision.

5. For fig 5F the plot the elution fractions of novel orfs to support the claim that they run higher than expected MW and are therefore in complex. But the monomer molecular weights are not shown on the plot so I think it cannot be interpreted in this way. Please add the monomer molecular weights (or expected fraction or whatever)

We added the predicted monomeric weight as red line in the corresponding heatmap (Fig 5F)

6. Proteomexchange access was not available for review

All data now is publicly available via the massive data repository with the identified MSV000091715.

signed
Ben Collins

Reviewer #2 (Remarks to the Author):

The manuscript "Next-generation interaction proteomics for quantitative Jumbophage-bacteria interaction mapping" is a very interesting piece of work and very much-needed research in the field. The work is quite impressive and very much accessible. The proteome work is very important to detect and determine the function unit of the phage and its host. The use of SEC-MS to study the proteome of the phage during their infection is very informative and valid study. Thus, the paper can be accepted in its native form. No further comments are recommended.

We thank the reviewer for the positive review.

Reviewer #3 (Remarks to the Author):

The manuscript from Fossati et al describes application of a new mass spectrometry method to analyse viral-host interactions, using *Pseudomonas aeruginosa* and two jumbophages, phiKZ and phiPA3.

There is an abundance of new data that appears thoroughly analysed. Known complexes are found as expected, which builds confidence in the approach, and new predictions are validated. Overall this should be of interest to those working in proteomics, phages, and viral-host interactions.

As a non-expert in MS, I will say that I found the manuscript rather opaque. The final impression is that thorough analysis has been performed, but is largely inaccessible to the generalist reader.

We thank the reviewer for this comment and acknowledge that we might have not provided enough information in some sections. To facilitate the readability of the manuscript, we added text to describe the following in detail.

1. General overview of SEC-MS experiment and expected outcomes (line 135-147)
 - a. *'In this technique, protein complexes extracted from a native lysate are size-fractionated and each fraction is analyzed via mass-spectrometry, resulting in a large matrix of protein intensities over molecular mass. These protein profiles are then utilized as a proxy for the assembly state under the assumption that proteins having identical peak shapes and position were physically associated at the separation stage.'*
2. Description of the differential SEC score and interpretation of peak position (line 527-530)
 - a. *'Differential analysis of two SEC profiles using PCprophet provides the SEC differential score which represents the variation of complex intensity (stoichiometry) or peak position (assembly state).'*
3. Scoring of SEC-MS datasets and utilization of computational tool and machine learning powered frameworks for complex prediction (line 234-246)
 - a. *'Following data processing and replicate integration, we utilized deep learning to predict co-eluting (i.e., interacting) proteins based on their intensity profiles across all measured fractions for a particular condition. To further increase our confidence we utilized two filters: first a target-decoy approach was employed to control for randomly coeluting proteins and then all PPIs having a false-discovery rate of less than 5% were filtered with a prediction probability greater or equal to 0.75'*

Specific comments:

Line 135 - can the authors tell us how similar the two phages are? What is the shared identity, are they taxonomically distinct?

Φ KZ (NC_004629.1) and Φ PA3 (NC_028999.1) share ~84% nucleotide sequence identity and an average of 64% of protein sequence similarity (see figure below). While many core proteins share homologs, they are often not identical (as exemplified by the plot attached), and several accessory genes are unique to one phage or the other. According to the most recent release of the International Committee on Taxonomy of Viruses (ICTV) the two bacteriophages are classified under the same class (*Caudoviricetes*) in separate genera, namely *Phikzvirus* (Φ KZ) and *Miltoncavirus* (Φ PA3).

Figure XX. Sequence similarity between phiPA3 proteins and phiKZ proteins. Only proteins with significant E-values are shown.

Line 166 - please briefly explain why 60 min was chosen

Line 166: We chose 60 minutes as a time point for SEC-MS for two main reasons:

1. It is approximately at the midpoint of the Jumbophage infection cycle (that lasts 90-120 min; ref Chaikerasitak, Science Advances, 2022), thus representing a sound starting point to study phage-host interaction complexes formed during infection.
2. At this time point, most known Φ KZ protein complexes are fully assembled and active, namely the Virion RNA polymerase (vRNAP), non-Virion RNA polymerase (nvRNAP), and the Shell. Thus we would expect to detect these complexes in our SEC-MS data set (which we indeed do).

Line 177 - why not do an uninfected control, or phage lysate control? Can you please state whether the KZ-resistant strain is also PA3 resistant?

Yes, Indeed the KZ-resistant strain is resistant to the bacteriophage PHIPA3. Please see the plaque assay attached below, which is now included as Supplementary Fig S1 of the manuscript.

The selection of a control is an important question. Arguably the best control is one that mimics the true experiment as similarly as possible. Here, the use of the receptorless bacteria (i.e., KZ-resistant strain) for our control experiments allows us to mimic the intracellular infection experiments as closely as possible, but without allowing for the formation of intracellular phage-bacteria assemblies. This control enables us to best

discriminate between bonafide intracellular infection assemblies and virion protein assemblies, which is important due to the large number (>50) of virion proteins in these phages. As such, we feel the use of additional uninfected controls or phage-lysate controls would not appreciably add any additional benefit or critical information for the interpretation and analysis of the SEC-MS data from the samples of phage-infected bacteria.

Figure XX. Plaque assays with Jumbophages Φ KZ and Φ PA3 spotted in 10-fold serial dilutions on a lawn of PAO1 (WT) or PAO1 (Φ KZ resistant mutant) strains. Clearings represent phage replication.

Line 263 - "majority of these single-peak proteins" can you please provide numbers.

We operatively defined a protein coeluting not at their expected molecular weight if the major peak is >4 fractions (>1 ml / at least 100 Kda MW shift). Based on this we further added the total number of single peak proteins (line 292).

'(90/137 single-peak proteins for *phiKZ*, 75/110 for *phiPA3* and 1843/2382 for the PAO1 control)'

Line 320 - what does a t-SNE plot show, is the shape significant in some way?

T-distributed stochastic embedding (t-SNE) is a non-linear approach for dimensionality reduction where the algorithm tries to maintain the local structure of the data (in other words, keep neighboring points close to each other) commonly employed in scRNAseq (<https://www.cancer.gov/ccg/blog/2020/interview-t-sne>) and which started to be utilized for high-dimensional proteomics such as spatial proteomics (PMID 27641956, 34800366). In contrast to traditional principal component analysis (PCA) where the distances between groups means increased distance in linear space, t-SNE distances and shapes are largely irrelevant, as the algorithm works by minimizing the distances between neighboring points rather than working by 'bulk' rotating vectors like a PCA.

We reasoned that this is a superior way of visualizing SEC data, given that PCA is highly sensitive to outliers (i.e proteins with high number of peaks or with choppy profiles), which could push close to each other proteins sharing similar shapes but not necessarily

predicted to be interacting. It is important to point out that any visualization (PCA, t-SNE, UMAP, SOM, etc) will lead to separation of vastly different protein profiles.

We added the following text at line (366-374) to clarify the aforementioned point.

'In this dimensionality reduction approach, neighboring points in the embedded space are derived from proteins sharing similar protein profiles and different profiles results in distant points, but due to the non-linear nature of the t-SNE algorithm, distance between clusters and the shape of the global or local embedding cannot be interpreted back to the input data.'

Figure 3 - I think I am correct in saying these data are comparing the phage+bug experiment vs the phage+resistant bug experiment. How are the two biological replicates handled? Are they represented here? How do the replicates compare?

Prior work showed the high reproducibility of SEC-MS measurements across replicates, especially when utilizing data-independent mass spectrometry (PMID 30642884, 24054720).

In our manuscript we also showed that the replicates are highly reproducible (Fig 1E) with an average r^2 of more than 0.8. Our software tool (PCprophet, PMID 33859439) leverages this utilizing the average between replicates to obtain a 'consensus' elution profile for a particular protein from the peptide-level data. Briefly, after selecting from each independent replicate for a specific condition the peptides showing the highest degree of correlation across the SEC dimension (i.e. sibling peptide correlation as defined in PMID 30642884), only peptides shared between all replicates are utilized for protein quantification. Missing values for a particular peptide are then imputed utilizing the correspondent measurements in the other replicates or during the curve fitting step.

Line 606 - I keep wondering about the use of "interactions". The experiment simply analyses the protein content of fractions from a SEC experiment. The proteins in the same fraction cannot be said to interact, they are simply co-eluting, and could be doing so away from expected molecular weight due to the abundance of stickiness, or co-factors, in those fractions, rather than a specific interaction. Doesn't there need to be adequate and cautious consideration of the truth of these interactions. Perhaps some cunning stats can be employed to provide confidence (and indeed perhaps has been, but escaped this reviewer).

We agree that we could have provided more information for this particular point. SEC-MS is built on the assumption that protein sharing identical peak profiles (i.e local peaks) were physically associated at the fractionation stage.

As an example in the attached figure, the orange protein is identified across many fractions but the coelution is only with the purple protein and not with the complex formed by the red and the green. So, specifically when we talk about interactions/coelutions between proteins

we refer to proteins sharing identical/close to identical peak shape in a region of the SEC dimension (i.e molecular weight/fraction number).

This implies that proteins identified in the same fractions but having different intensity profiles (i.e peak shapes, like the orange protein in respect to the green/red one) across all 72 fractions are not interacting. Clearly, for this guilt-by-association principle to work there must be a scoring system to robustly separate spuriously co-eluting/co-identified proteins and real interactors prior to claiming interacting proteins. Recently, we introduced a machine-learning powered toolkit, which was trained using several thousands protein coelution profiles and we demonstrated that it can effectively predict protein complexes with high accuracy from co-fractionation MS datasets (PCprophet, PMID 33859439). We further introduced a deep learning framework dubbed Protein-Protein Interaction Prophet (PPIprophet, doi

<https://doi.org/10.1101/2023.03.22.533843>) which was trained on >1,000,000 PPIs from 30 different co-fractionation datasets (encompassing different MS acquisition schemes, separation chemistries and number of fractions) which we showed being particularly effective for detection of low-abundant PPIs. All of our interactions are predicted with confidence/probability >0.75 from the PPIprophet toolkit, which means some low-confidence (hopefully false positives) but positively predicted interactions are removed. It is important to note that, as the reviewer correctly pointed out, there are many factors on why proteins might coelute together like aggregates, stickiness of particular proteins, yet unidentified complex subunits etc. While we do our best to control for the experimental factors responsible for some of these factors (i.e avoid too high protein concentrations, ultracentrifuge prior to SEC etc) as well as utilized advanced ML/DL models to predict interacting proteins using heterogeneous data sources, full utilization of SEC profiles is still an open topic and active area of research to improve confidence in the identified complexes and further reduce false positive coelutions. To summarize this point, we added the following line to the result section to explain the scoring process (line 234-246).

Following data processing and replicate integration, we utilized deep learning to predict co-eluting (i.e interacting) proteins based on their intensity profiles across all measured fractions for a particular condition. To further increase our confidence we utilized two filters, first a target-decoy approach was employed to control for randomly coeluting proteins and

then all PPIs having a false-discovery rate of less than 5% were further filtered with a prediction probability of > 0.75'

Furthermore, we added this line to the discussion to emphasize the possibility of false positive interactions in our data (line 1306-1310)

'This step, while increasing the confidence in our PPIs identification, still does not allow for the complete removal of false-positive results due to the lack of a real ground-truth dataset, which is to be expected in large-scale fractionation experiments.'

Line 609 - following on from above, these numbers are surprising. "292 interactions between pairs of KZ viral proteins and 6550 host-pathogen between KZ and PA01 proteins". Is this saying 292 interactions between phage proteins (believable) and 6550 interactions phage+host? Isn't 6550 astonishingly high? You say earlier (line 240) you detected 280 KZ proteins. $6550/280 \sim 23.4$. Do these data say each phage protein on average interacts with 23.4 host proteins? That doesn't seem reasonable. Is there precedent for this large an interaction network, or is this simply co-elution without true biological interaction? Can these numbers be justified as reasonable, to bat away doubt that a lot of the "interactions" are actually just noise?

The reviewer brings up a good point that is important to clarify how interactions are counted in different types of experiments. For an APMS experiment where one purifies a given tagged protein of interest, the interactions detected are counted in a binary fashion between the protein of interest (e.g. Protein A) and all other proteins detected (e.g. A-B, A-C, A-D, etc.). In such an experiment one is unable to detect higher level protein complexes to know if protein C and D interact with one another. However, SEC-MS is a protein complex interaction analysis method. Such that if we find two coeluting proteins (Protein A and B), then we can assign a single interaction (A-B). However, if we find three proteins coeluting (Protein A, B, C), we assume they form a complex and count every pairwise interaction possible (A-B, A-C, B-D). As you might imagine, when scaling this reasoning to protein complexes with a diversity of sizes and across an entire proteome, it is not unforeseeable that we would obtain a large number of interactions, of which many are likely derived from coelution of proteins within large assemblies. As mentioned in the previous reply, we further acknowledge that there is a certain percentage of false positive interactions. In our scoring toolkit we have a target-decoy competition scoring which we utilized for false-discovery rate control (fixed at 5%) so the interactions are pre-filtered prior to the 0.75 subsetting.

Line 737 - Really good to see separate validation of interactions with the ribosome. Nothing negative here - I think it's a lovely experiment.

We thank the reviewer for the positive feedback.

Typos lines 109, 165, 704, 733, 734, 769

We corrected the typos and thank the reviewer for pointing them out.

REVIEWERS' COMMENTS

Reviewer #1 (Remarks to the Author):

All comments addressed well. Recommend accept.

Reviewer #3 (Remarks to the Author):

The authors have done an excellent job of addressing this Reviewer's comments. I thank them for taking the time to clearly explain the fine details. I congratulate them on their study and suggest no further changes.